# Genome-wide quantitative dissection of an arthropod segmented body plan at single-cell resolution
Takanori Akaiwa[1,2,5], Hiroki Oda [1,2] & Yasuko Akiyama-Oda [1,3,4] ✉

Developmental processes underlying the characteristic segmented body plans in arthropods vary widely. While *Drosophila* is well-studied, few other arthropod species offer platforms for comparable genomics at single-cell resolution. Here, we present high-quality quantitative data from single-nucleus RNA sequencing of spider *Parasteatoda tepidariorum* embryos at late stage 5 and stage 7, a critical period of emergence of segmental units along the anterior–posterior (AP) axis. Clustering analysis of the stage-7 dataset reconstructs an axial alignment of ectoderm cells, reflecting the differing cell states along the segmenting AP axis. This enables us to obtain genome-wide quantitative gene expression profiles along the reconstructed axis, which were used for unbiased and thorough molecular investigation of pattern elements employing statistical methods. Comprehensive gene-to-gene correlation analyses suggest distinct gene-regulatory interactions in different regions along the reconstructed axis. This study lays the foundation for exploring the origins of developmental diversity in the arthropod body plan.

Multicellular animals within the same phyletic group share characteristic features; however, the developmental processes that generate these features may be remarkably diverse. A prime example of this is the segmented body plan of arthropods, which is a conserved trait achieved through diverse cellular and molecular processes across species[1,2]. Understanding how such shared traits originated and diversified during evolution, particularly from a genomic perspective, is a crucial objective in evolutionary and developmental biology.

Segmentation along the anterior–posterior (AP) axis is one of the most well-known biological processes that has provided the opportunity for exploring the mechanisms of patterning from experimental and theoretical perspectives[3–14]. Genetic studies on segmentation in the *Drosophila melanogaster* embryo through comprehensive identification of the genes involved and their interactions have provided a foundational understanding of the mechanisms underlying arthropod segmentation[15–17]. Candidate-gene approaches, focusing on orthologs of *Drosophila* segmentation genes, have expanded the knowledge on segmentation to a wider range of arthropod species[18–25]. However, these approaches have revealed substantial variations in mechanisms and processes, despite the conserved outcome of periodic gene expression, such as the expression of *engrailed* in one stripe per segment. Genome studies have also shown variations in gene content

even within the same phyla[26–30]. To understand how animals, as genetic and cellular systems, could have diversified the processes of segmentation, model platforms are required for collecting and integrating the genome-wide, unbiased, cell-level, and quantitative data suitable for comparative studies.

The spider *Parasteatoda tepidariorum* offers a valuable alternative model system to *Drosophila*. Unlike the syncytial environment of the *Drosophila* embryo, where the transcription factor Bicoid plays a crucial role in establishing the global AP polarity[31,32], patterning in the *Parasteatoda* embryo predominantly occurs in a cellular environment[33,34], with the cell-cell signaling protein Hedgehog playing a similar role[35]. In the downstream patterning processes, segmental stripes arise via temporally repeated expression of genes in *Parasteatoda*[21,25,36–38], in contrast to spatial regulation by region-specifically expressed gap genes through stripe-specific enhancers in *Drosophila*[39–41]. The dynamics of the temporally changing expression in *Parasteatoda* vary across body regions: bi-splitting in the head and oscillation in the posterior region, together with a non-temporally repeated dynamics, that is, tri-splitting in the thorax[21,25,38,42]. Furthermore, genes such as *Pt-msx1*, which have no known role in *Drosophila* segmentation, contribute to these dynamics of *Parasteatoda* stripe formation[38]. These molecular differences within the conserved patterning outcomes underscore the developmental diversity underlying the shared segmented body plan. With a

[1]JT Biohistory Research Hall, Takatsuki, Osaka, Japan. [2]Department of Biological Sciences, Graduate School of Science, Osaka University, Toyonaka, Osaka, Japan. [3]Department of Microbiology and Infection Control, Faculty of Medicine, Osaka Medical and Pharmaceutical University, Takatsuki, Osaka, Japan. [4]PRESTO, Japan Science and Technology Agency, Kawaguchi, Saitama, Japan. [5]Present address: Exploratory Research Center on Life and Living Systems (ExCELLS), National Institutes of Natural Sciences, Okazaki, Aichi, Japan. ✉e-mail: yasuko@brh.co.jp

well-annotated, chromosome-level genome assembly, available experimental tools and techniques, bioinformatics resources, and a developing mathematical modeling platform, *Parasteatoda* provides an excellent platform for in-depth investigation of the formation of the stripe pattern[43–47]. Given the importance of temporal transcriptional dynamics within each cell in *Parasteatoda* segmentation and the distinct molecular mechanisms involved, a single-cell-level genome-wide analysis is crucial for understanding the patterning processes.

Single-cell and single-nucleus RNA-sequencing (scRNA-seq and snRNA-seq, respectively) are powerful techniques that enable genome-wide profiling of transcriptional states in individual cells, providing valuable insights into the trajectories of cell differentiation during animal development[48–57]. However, integrating the spatial information with data obtained from dissociated cells and nuclei remains a challenge, despite the development of several integration techniques[55,58–61]. Our group and others have applied scRNA-seq and snRNA-seq techniques to *Parasteatoda* embryos to unravel the cell clusters associated with body regions and specific differentiation trajectories[62–64]. Furthermore, the previous data derived from a small number of sibling embryos at late stage 5 have successfully reconstructed the alignment of cells along the radius of the germ disc (the future AP axis) in a Uniform Manifold Approximation and Projection (UMAP) plot, reflecting gradually different transcriptional states[62]. These findings highlight the unique opportunity presented by the *Parasteatoda* embryo for leveraging scRNA-seq and snRNA-seq to comprehensively investigate spatial gene expression patterns, particularly those reflecting the AP polarity, and gain insights into the molecular mechanisms underlying the body axis patterning.

In this study, we expanded single-cell analysis to later-stage spider embryos exhibiting more complicated stripe patterns. We demonstrate that data obtained from snRNA-seq contained sufficient information to reconstruct the cellular states along the AP axis, accurately reproducing the observed stripe pattern. This reconstructed pattern was then used to generate quantitative gene expression profiles, which were subsequently utilized to identify gene groups exhibiting similar stripe patterns. Furthermore, novel cell states within the mesoderm, endoderm, and extraembryonic tissues were identified. This study establishes a novel platform for conducting genome-wide, single-cell-level, and quantitative analyses of the patterning processes underlying the shared features of arthropods.

## Results

### Single-cell and single-nucleus RNA-seq of germ-band cells

We conducted scRNA-seq and snRNA-seq using *Parasteatoda* embryos at stage 7 [51 h after egg laying (AEL)] in essentially the same way as previously described for embryos at late stage 5 (39 h AEL)[62] and analyzed the late stage-5 and stage-7 datasets together. There is a key transition associated with axis formation and segmentation. The late stage-5 germ disc, which consists of approximately 2000 cells, represents the radial symmetry of gene expression with AP polarity in the peripheral to central order, whereas the stage-7 nascent germ band, which consists of approximately 3500 cells, represents repetitive and nonrepetitive stripes of gene expression along the AP axis (Fig. 1A)[25,38].

We constructed stage-7 scRNA-seq and snRNA-seq libraries using 15 and 20 carefully staged embryos, respectively (Fig. 1B; Supplementary Fig. 1) and obtained reliable data from 865 cells and 6239 nuclei by filtering out data stemming from potential doublets or low-quality cells and nuclei (Table 1; Supplementary Fig. 2, Supplementary Data 1). The relatively low recovery rate for cells in the scRNA-seq experiment was due to technical difficulties associated with cell dissociation and sedimentation during library construction. To record the precise developmental stages of the embryos used for library construction in the context of stripe-pattern formation, sibling embryos were simultaneously fixed and stained for *Pt-hh* and *Pt-msx1* transcripts (Supplementary Fig. 1). All five embryos from each sample were at very similar stages, displaying bi-splitting stripes in the head, tri-splitting stripes in the thorax, and stripes formed via oscillatory gene expression in the posterior region. Based on the gene expression pattern, the

stages of embryos used for the single-cell preparation were slightly younger than those used for the single-nucleus preparation.

Clustering analysis was performed against the cells and nuclei from the three datasets (late stage-5_nucleus, stage-7_nucleus, and stage-7_cell) using the Seurat package with various parameter settings. The representative results are displayed as colored dot plots in the 2-dimensional (2-D) space using UMAP, characterizing the cell states, and as violin plots characterizing the clustered cell populations (Figs. 1C–I, 2; Supplementary Figs. 3–5). These UMAP plots similarly exhibited the spatial separation of cells/nuclei into one large group and several smaller groups, irrespective of the dimension values used in the analyses (Supplementary Fig. 3). Here, the term "supercluster" of cells/nuclei is used to indicate a group of cells/nuclei plotted in a close area of the 2-D space of the UMAP throughout this study. Each supercluster consisted of multiple clusters and their numbers changed markedly depending on the resolution parameters (Fig. 1I). The mean number of genes detected per cell/nucleus in the clusters ranged between 2449 and 3010 in the late stage-5_nucleus sample (except for cluster 9), between 3457 and 4125 in the stage-7_nucleus sample (except for cluster 3), and between 2499 and 2827 in the stage-7_cell sample (Fig. 1F–H). Only in the late stage-5_nucleus cluster 9 and in the stage-7_nucleus cluster 3 were the gene numbers lower than those in the other clusters, and the same trend was observed for the RNA counts. Furthermore, cluster markers for these and derived clusters were not specifically expressed, showing higher expression levels in other clusters or very faint expression in the respective clusters (Fig. 2, clusters 3, 13, and 15; Supplementary Fig. 4C). It is possible that these two clusters were pseudoclusters and were, therefore, excluded from further analyses.

The clusters of cells/nuclei in the UMAP plots were characterized by the expression of known marker genes (Table 2). The largest late stage-5_nucleus supercluster consisted of the presumptive ectoderm, mesoderm (pMES and cMES cells), and extraembryonic cells, with a recapitulated AP order of gene expression domains observed in the germ disc (Fig. 1C) as previously reported[62]. The dorsal-inducing CM cells and endoderm cells formed a supercluster but were sharply separated from each other (Fig. 1C; Supplementary Fig. 4D). The largest stage-7_nucleus supercluster consisted of only ectoderm cells, showing an elongated area of distribution with a recapitulated order of gene expression domains corresponding to the head-thorax-opisthosoma sequence (Fig. 1D). The higher the resolution, the more clusters emerged along the longitudinal axis of the supercluster (Fig. 1D, I). The largest stage-7_cell supercluster consisted of only ectoderm cells with distinct clusters corresponding to the head, thorax, and opisthosoma but showed a less elongated distribution (Fig. 1E). No axial alignment of the ectoderm clusters was observed when the resolution was increased (Supplementary Fig. 5A). Two separate mesoderm superclusters were identified in the nucleus sample at stage 7 (Fig. 1D), whereas in the cell sample there were two separate mesoderm clusters (Fig. 1E), which were not subdivided into smaller clusters in clustering with higher resolution values (Supplementary Fig. 5A). Using an endoderm/extraembryonic cell marker (*At_eW_012_A08*)[36], two superclusters were identified in the single-nucleus sample (Fig. 1D) and one cluster in the single-cell sample (Fig. 1E). Overall, these results indicate that the cell states for the germ layers are well separated by stage 7 (51 h AEL), and that the axial alignment of the ectoderm clusters occurs in the 2-D space of the UMAP when using the nuclei dissociated from stage-7 embryos.

### Characterization of the axial alignment of ectoderm clusters in the single-nucleus UMAP plots

Next, we characterized the ectoderm cell array in the stage-7_nucleus UMAP plots constructed at serial resolutions. Three, five, eight, eleven, and thirteen clusters were aligned along the axis at resolutions of 0.1, 0.5, 1.0, 2.5, and 3.3 plots, respectively (Fig. 1D, I). Of these resolutions, the resolution of 2.5 (Fig. 1I, the third plot and Fig. 2) was used for the comprehensive characterization of cell states, as the cluster number was comparable to the number of segmental units forming at stage 7. We identified marker genes for the ectoderm cell clusters at stage 7 (Supplementary Data 2) and stained

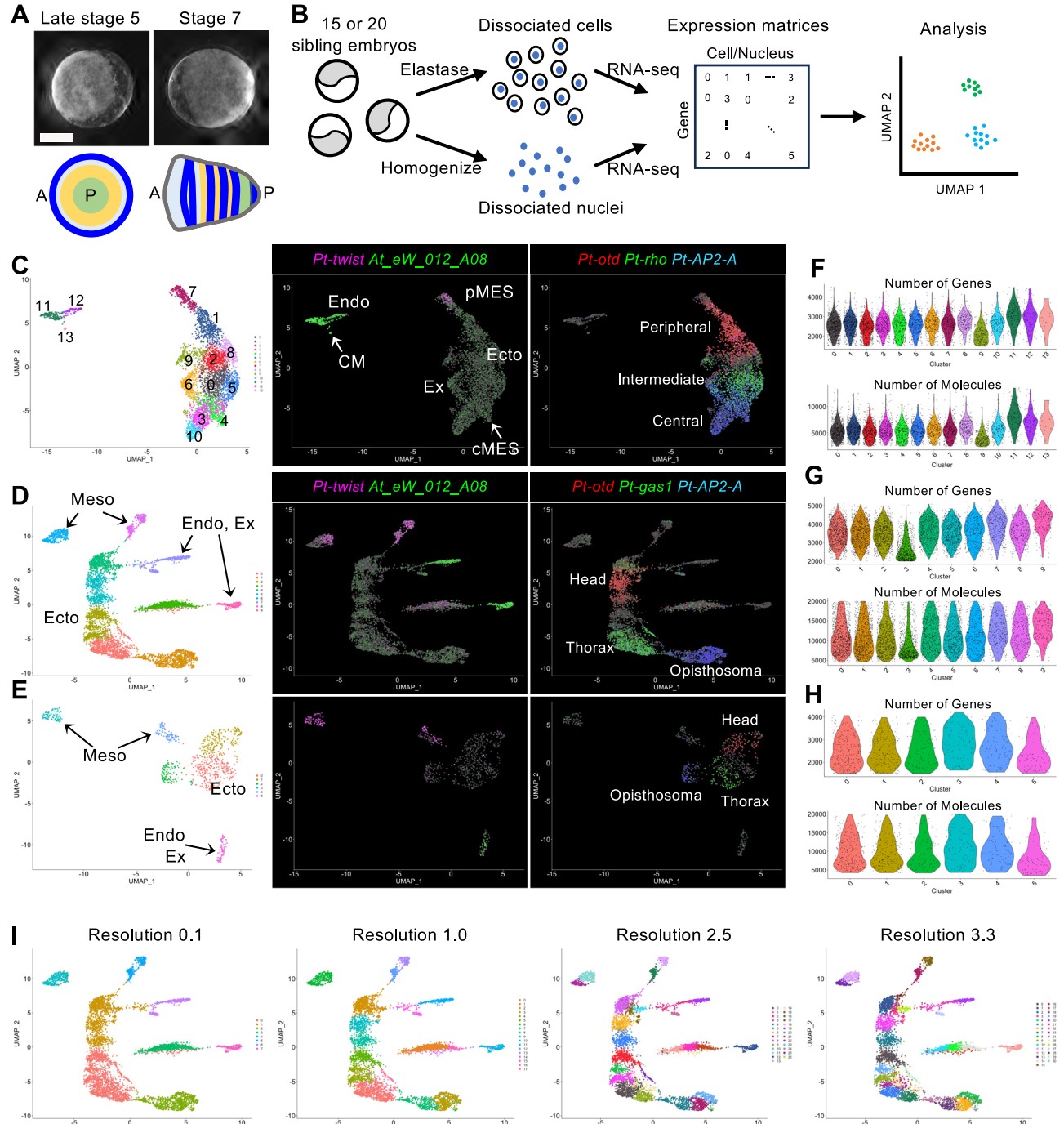

**Fig. 1 | Single-cell and single-nucleus analysis. A** Spider embryos at late stage 5 (39 h AEL) and stage 7 (51 h AEL). Microscopic images (top) and schematic illustrations showing characteristics of gene expression patterns in the germ disc and germ band at these stages (bottom). Bar = 200 μm. **B** Illustration of scRNA-seq and snRNA-seq protocols. **C–E** UMAP plots of the late stage-5_nucleus (**C**), stage-7_nucleus (**D**), and stage-7_cell (**E**) RNA-seq data. The clustering parameters of the dimension and the resolution are 1:45 and 1.4 (**C**), 1:50 and 0.5 (**D**), and 1:50 and 0.5 (**E**). Middle and right panels show the expression of mesodermal and endodermal markers (middle) and body region markers (right). Gene names and colors are indicated. Endo endoderm, Ecto ectoderm, Meso mesoderm, pMES peripheral mesoderm, cMES central mesoderm, Ex extraembryonic tissue, CM cumulus mesenchymal cells. **F–H** Violin plots showing the number of genes (upper) and molecules (lower) detected per cell or nucleus of the clusters in the UMAP shown in (**C–E**). **I** Stage-7_nucleus UMAP plots constructed with multiple resolution parameters, 0.1, 1.0, 2.5, and 3.3, at a dimension of 1:50.

the embryos for the transcripts of several selected markers (Fig. 3A). The staining revealed a perfect correlation between the AP positions of the gene expression domains in the germ band and the positions of the expressing cell population in the ectoderm cell array of the UMAP plot. This included the characteristic expression pattern of LOC107451631 (*Pt-osr-B*) in the two distinct domains. These observations indicate that the global polarity of the

embryonic AP axis in the ectoderm layer was reconstructed in the stage-7_nucleus UMAP plot.

The aligned clusters of the stage-7_nucleus and late stage-5_nucleus datasets were examined and compared using the expression of the top-10 markers for each of the eleven stage-7 clusters and each of the seven late stage-5 clusters (Fig. 3B–G; Supplementary Data 2, 3). The late stage-5

## Table 1 | Summary of scRNA-seq and snRNA-seq

| Library | Stage-7 cell | Stage-7 nucleus | Late stage-5 nucleus |
|---|---|---|---|
| No. of embryos | 15 | 20 | 20 |
| No. of cells | 865 | 6239 | 4820 |
| Ave. genes/cell | 2553 | 3456 | 2558 |
| Ave. molecules/cell | 9988 | 11,020 | 5561 |
| Total genes | 13,766 | 19,692 | 19,099 |

marker gene set revealed gradually differing cell states along the peripheral-central alignment of the late stage-5 germ disc (Fig. 3D)[62]. However, the same set of genes exhibited the position-dependent cell states less clearly in the stage-7 ectoderm, showing expression in multiple ectoderm clusters (Fig. 3E). Similarly, most of the stage-7 markers showed the highest expression levels in the respective clusters, but the levels were lower in the neighboring clusters as well as in more distant ones in the stage-7 data (Fig. 3G), which manifests gradually differing cell states along the axis. In the late stage-5 data, similar tendencies were recognizable in the expression of these markers but were much less prominent (Fig. 3F), with expression of some of them (e.g., markers of clusters 1, 6, 0, 7, and 10) showing very weak expression at late stage 5. These results indicate that the cell states at stage 7 are derived from those at late stage 5, and that newly expressed genes contribute to specifying the varied cell states during the development. In addition, the cell states at and near the stage-7 anterior and posterior termini (clusters 1/6 and clusters 4/7/10) were characterized by a relatively restricted expression of marker genes.

In contrast to the nucleus plots, the stage-7_cell UMAP plot exhibited a less organized pattern of cell populations expressing the AP markers (Supplementary Fig. 5B, compared with Fig. 3A). The number of nuclei used for the stage-7_nucleus plot ($n = 6239$) was much larger than that used for the stage-7_cell plot ($n = 865$) (Table 1). To assess the effect of data volume on pattern reconstruction, we prepared 10 subsets of randomly selected 865 nuclei from the stage-7_nucleus data and analyzed them for clustering. The resulting UMAP plots showed axially aligned ectoderm clusters in seven cases (trials 1–6, 8), but less prominent in the remaining cases (Supplementary Fig. 6). We could not conclude whether the inability to reconstruct the AP pattern using the stage-7_cell dataset was due to insufficient data volume, the property of data derived from the cytoplasm, or both.

### Characterization of the mesoderm and endoderm clusters

We next characterized the cell states of the non-ectoderm clusters in detail using known and newly identified markers (Fig. 4; Supplementary Figs. 4, 7, 8). The late stage-5_nucleus clusters 11, 12, and 13 formed the endoderm supercluster (Figs. 1C, 4A), and of these, cluster 13 was the CM cells (Supplementary Fig. 4D), as mentioned above. LOC107443634 (*aug3.g486*)[65] marked only cluster 11, whereas LOC107436131 (the present study) marked both clusters 11 and 12 (Supplementary Fig. 4E). Multicolor fluorescence in situ hybridization (FISH) for these two genes revealed two cell states of endoderm cell populations beneath the germ disc (Supplementary Fig. 4F). The locations of the marked cells indicate that this difference in the cell state is not relevant to the classification of pEND and cEND cells, which were named after their birthplace (i.e., the peripheral and central sides of the germ disc[36]). The cells located on the surface and outside the germ disc area were also included in cluster 11 (Fig. 4A, B). The top-10 markers of cluster 12 were expressed not only in the cluster-12 cells but also in the cells of clusters 11 and 13, in contrast to the markers of clusters 11 and 13, which showed more specific expression (Supplementary Figs. 4C, 7C).

The stage-7_nucleus clusters 12, 19, 21, and 24 were endoderm and extraembryonic cells (Figs. 1D, 2). Staining of embryos with specific markers revealed that the cells of cluster 12 were typical endoderm cells located between the surface layer and yolk mass, with a scattered distribution over the embryo (Fig. 4C, D; Supplementary Fig. 8A, B). This cluster was related to the late stage-5_nucleus cluster 11, as was evident from the expression of

the top-10 markers (Supplementary Fig. 7D, E). The stage-7_nucleus cluster 19 showed a slight similarity to the late stage-5_nucleus endoderm clusters (clusters 11 and 12) in terms of marker expression (Supplementary Fig. 7D); however, staining of a specific marker revealed extraembryonic cells (Supplementary Fig. 8C). The stage-7_nucleus clusters 21 and 24, forming a supercluster with the cluster 19, were determined to correspond to surface cells at and near the boundary between the embryonic and extraembryonic regions; cluster-24 cells were at the caudal terminal area and cluster-21 cells were at a more anterior area (Fig. 4D; Supplementary Fig. 8C, D). The cell states of these two clusters were similar to those of late stage-5_nucleus cluster 6 (Supplementary Fig. 7D), which were emerging extraembryonic cells (Supplementary Fig. 4D).

Two mesoderm superclusters in the stage-7_nucleus UMAP consisted of two clusters each, namely clusters 16/18 and clusters 11/22 (Figs. 1D, 2). Staining of the marker gene expression revealed four subpopulations of the mesoderm, defined by the cell states of the clusters 16, 18, 11, and 22, in this order, from the anterior to the posterior (Fig. 4D; Supplementary Fig. 8E, F). The cell states of clusters 16 and 18 were relatively similar to those of late stage-5_nucleus cluster 7, and the cell states of clusters 11 and 22 were relatively similar to those of late stage-5_nucleus cluster 4 (Supplementary Fig. 7D), consistent with the previous descriptions of pMES and cMES cell development[36].

### Detection of emerging segmental units in the reconstructed stage-7 ectoderm

As described above, the stage-7_nucleus dataset reconstructed the axial alignment of the ectoderm nuclei (Fig. 3). Next, we examined whether the stage-7 datasets could also reconstruct the repeated stripes of gene expression associated with segmentation in the plots. To address this question, we visualized the cell populations expressing *Pt-msx1*, *Pt-hh*, and *Pt-noto1* in the stage-7_nucleus plot in comparison to the same-stage embryos stained for their transcripts (Fig. 5A, B). This comparison showed that the periodic stripe pattern in the embryo was recapitulated in the plot, where the complementary patterns of *Pt-hh* and *Pt-msx1,* as well as those of *Pt-noto1* and *Pt-msx1* were observed along the axis. Notably, patterns corresponding to the bi-splitting stripes in the head, tri-splitting stripes in the thorax, and stripes formed by oscillatory expression in the posterior region were observed in the UMAP plot. In contrast, the stage-7_cell data did not exhibit any segmental patterns (Fig. 5C).

### Generation of genome-wide gene expression profiles along the AP axis in the ectoderm

The reconstruction of the axial alignment of the ectoderm using the stage-7_nucleus dataset prompted us to predict gene expression profiles in a genome-wide and quantitative manner. To this end, we developed a method for extracting gene expression levels along the reconstructed AP axis (Fig. 5D–G; Supplementary Fig. 9A). In this method, a spline curve running along the axial alignment of the ectodermal nuclei was set using a contour map of the nucleus density (Fig. 5D, E), and the distance between each nucleus and the spline curve was calculated (Supplementary Data 4). The expression levels of each gene in the nuclei within a threshold distance (Fig. 5E) were plotted against the position along the reconstructed axis, and the plotted data were further smoothened (Fig. 5F, G) to obtain a one-dimensional expression profile for each gene in the genome (Fig. 5H, I; Supplementary Data 5). The expression profile comprised the values at 80 subdivisions of the reconstructed axis, starting from the anterior end of cluster 1 to the posterior end of cluster 10, which were defined by the expression of *Pt-six3-1* and *Pt-prd2* (Supplementary Fig. 9A). The distant range used for selecting nuclei along the spline curve was optimized so that the generated expression profiles reflected real gene expression patterns. The generated expression profiles of *Pt-hh*, *Pt-msx1*, and *Pt-noto1* represented the characteristic peak patterns observed in the head, thorax, and opisthosoma (Fig. 5H). Further comparisons also supported the fact that the obtained expression profiles reproduced the complex patterns of gene expression along the AP axis in detail (Fig. 5I; Supplementary Fig. 9B).

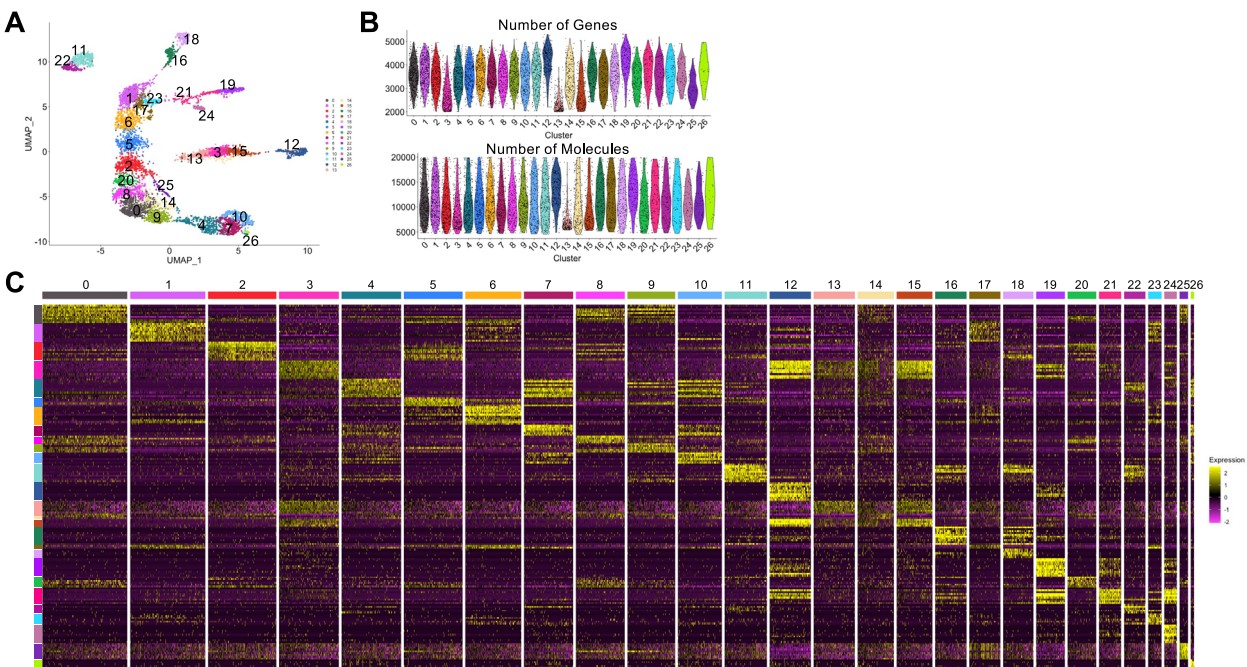

**Fig. 2 | Stage-7_nucleus clusters. A** UMAP plot of stage-7_nucleus RNA-seq data generated with a dimension of 1:50 and a resolution of 2.5. **B** Violin plots showing the number of genes (upper) and molecules (lower) detected per nucleus of the clusters in (**A**). **C** Heatmap showing expression levels (log-normalized) of markers in nuclei of each cluster. Top-10 markers of each cluster are shown, but those that have already appeared in earlier clusters are not shown.

**Table 2 | Marker genes used for the identification of cell types**

| NCBI ID | Gene name | Late stage 5 | Stage 7 | Ref. |
|---|---|---|---|---|
| LOC107440133 | *Pt-twist* | pMES | Mesoderm | 36, 80 |
| LOC107451427 | *At_eW_012_A08* | Endoderm, CM cells | Endoderm, extraembryonic cells | 36 |
| LOC107456525 | *Pt-Delta* | pEND, cMES | Stripes | 36 |
| LOC107451717 | *Pt-ets4* | CM cells | | 65, 81 |
| LOC107444265 | *Pt-noggin-D* | Extraembryonic region | | 38 |
| LOC107457189 | *Pt-otd* | Periphery | Head | 72 |
| LOC107438948 | *Pt-rhomboid* | Intermediate region | | 38 |
| LOC107438015 | *Pt-gas1* | Center | Thorax | 38 |
| LOC107452006 | *Pt-AP2-A* | Center | Opisthosoma | 38 |
| LOC107451809 | *Pt-hh* | Periphery | Stripes | 21, 36 |
| LOC107447678 | *Pt-msx1* | Center | Stripes | 38 |
| LOC107447988 | *Pt-noto1* | Broad | Stripes | 25 |
| LOC107450741 | *Pt-six3-1* | | Anterior end | 21 |
| LOC107445228 | *Pt-prd2* | Center, pMES | Stripes, posterior | 38 |
| LOC107452883 | *Pt-aslH* | Posterior end | Stripes, posterior | 38 |

The same method was applied to the late stage-5_nucleus dataset to obtain one-dimensional expression profiles along the radius of the germ disc, from the peripheral end of cluster 7 to the central end of cluster 10, which were defined by the expression of *Pt-twist* and *Pt-aslH* (Supplementary Fig. 9C, D). The late stage-5 expression profile also comprised values at 80 subdivisions, including the presumptive ectoderm and mesoderm. We confirmed that the predicted expression profiles of several selected genes recapitulated the actual expression patterns along the radius of the germ disc (Fig. 5J; Supplementary Fig. 9E, Supplementary Data 6, 7). The genome-wide gene expression profiles predicted from the late stage-5_nucleus and stage-7_nucleus datasets are available in our database and in the figshare repository[66,67].

**Genome-wide search for genes expressed in specific patterns**

The quantitative expression profiles obtained allowed us to use a statistical method to perform a genome-wide search for genes that are expressed in specific patterns along the AP axis. We performed two rounds of hierarchical clustering of all genes expressed in the ectoderm supercluster using the mean and difference between the maximum and minimum values collected from the expression profiles (Fig. 6A). In the first round of clustering of 16,023 genes using the stage-7_nucleus dataset, the genes were divided into 12 groups (Supplementary Fig. 10). From these, five groups were selected based on the following criteria: relatively small mean values and relatively large differences. In the second round, each group was further classified into two to five subgroups (Supplementary Data 8). Eleven

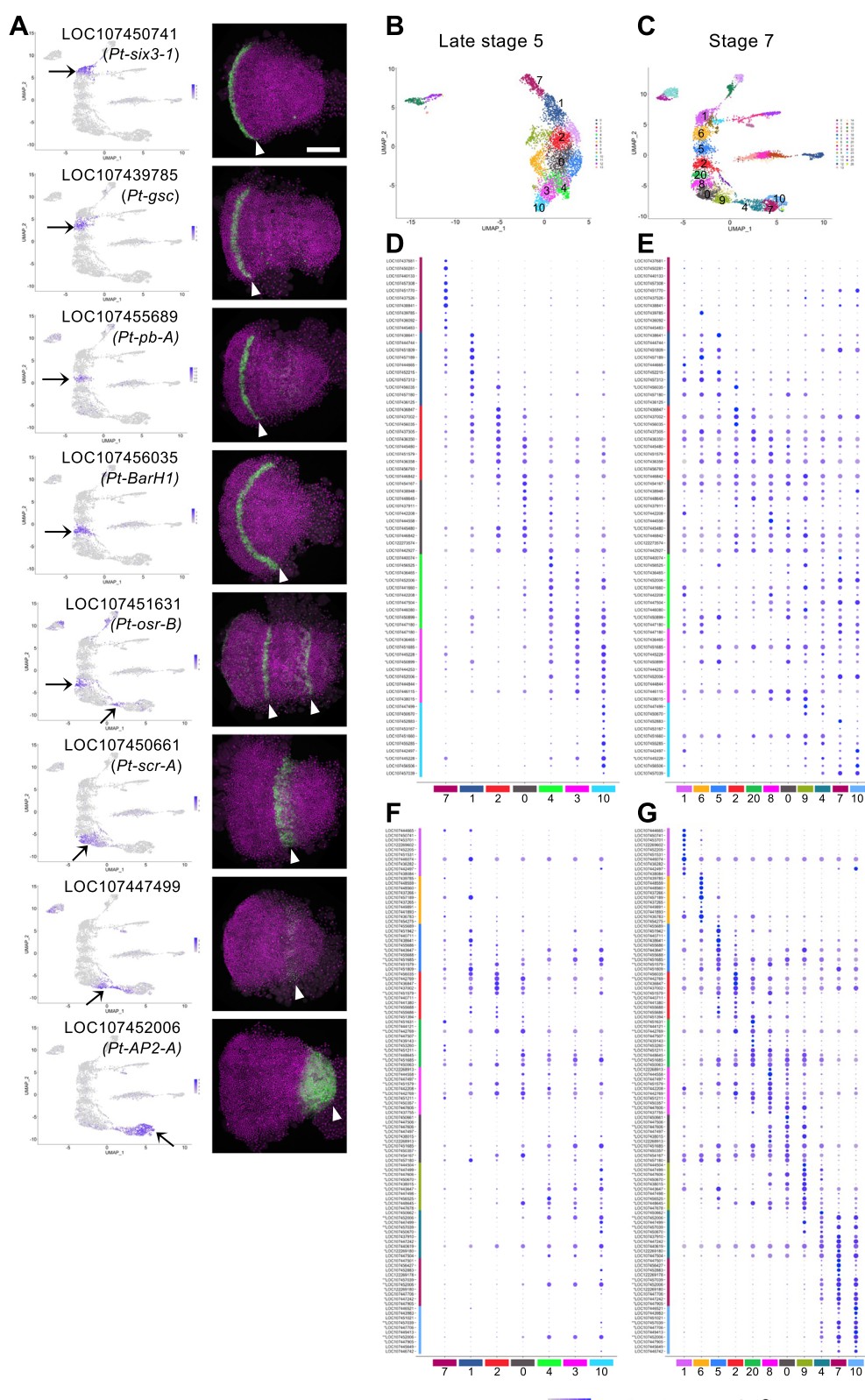

**Fig. 3 | Expression of stage-7 ectoderm cluster markers. A** Expression of stage-7 ectoderm cluster markers in the UMAP plots (left) and in stage-7 embryos detected using FISH (green, right). All embryos were counterstained with DAPI (purple). Arrows point to expression domains in the plot, and arrowheads to those in the embryo. Scale bar = 200 μm. **B**, **C** UMAP plots of late stage-5_ (**B**) and stage-7_ (**C**) nucleus data. The clusters aligned along the reconstructed AP axis and examined in the following analysis are indicated by numbers. **D**–**G** Dot plots showing the

expression of top-10 markers of the late stage-5 ectoderm and mesoderm clusters 7, 1, 2, 0, 4, 3, and 10 (**D**, **E**) and those of the stage-7 ectoderm clusters 1, 6, 5, 2, 20, 8, 0, 9, 4, 7, and 10 (**F**, **G**) in the late stage-5 (**D**, **F**) and stage-7 (**E**, **G**) clusters. Average expression levels (log-normalized) and percent expressed in the nuclei in each cluster are shown by the colors and sizes of the circles. Asterisks indicate genes that appear multiple times in the list (* twice, ** three times).

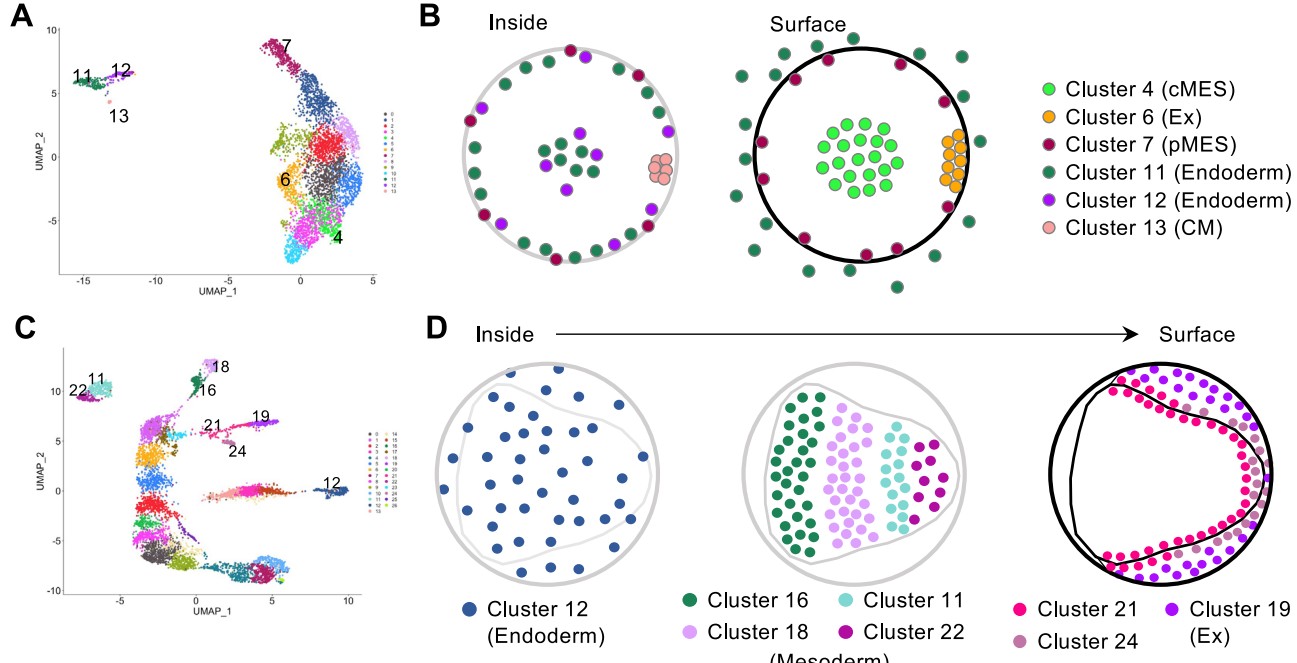

**Fig. 4 | Summary of mesoderm and endoderm clusters. A** Late stage-5 UMAP showing the position of the non-ectoderm clusters. **B** Schematic illustrations of the inside and surface layers of the late stage-5 germ disc showing cluster-11 and cluster-12 endoderm cells, cluster-7 pMES cells, cluster-4 cMES cells, cluster-13 CM cells, and cluster-6 extraembryonic cells. **C** Stage-7 UMAP showing the position of the non-ectoderm clusters. **D** Schematic illustrations of the inside-to-surface layers of the stage-7 germ band showing cluster-12 endoderm cells, clusters-16, 18, 11, and 22 mesoderm cells, and clusters-21, 24, and 19 cells from the periphery of the embryo to the extraembryonic tissue.

subgroups were selected based on the same criteria (Fig. 6B). A total of 213 genes were selected (Supplementary Data 9). Although this simple clustering-based gene selection worked well, some genes known to be expressed with specific patterns along the AP axis in the stage-7 germ band [e.g., LOC107453167 (*Pt-krü-1*)] were not included in the 213 genes. These genes exhibited very low expression counts in the dataset.

Similar hierarchical clustering analyses were performed on the late stage-5 expression profiles of 15,061 genes, and ten subgroups containing 264 genes were selected (Supplementary Fig. 11, Supplementary Data 10, 11). The expression profiles of the 213 selected genes at stage 7 and 264 genes at late stage 5 showed a variety of patterns (Fig. 6B; Supplementary Fig. 11C).

**Reconstruction of the AP pattern using the 213 selected genes**
To examine the contribution of the selected genes to reconstructing the AP pattern, we prepared two submatrices from the normalized stage-7_nucleus matrix. One submatrix was composed of 4102 ectoderm nuclei and the 213 genes identified in the above analysis (submatrix A), and the other was composed of the same nuclei and the full gene set, but excluding the 213 genes (submatrix B). Seurat analysis using submatrix A reconstructed the axial alignment of the nuclei, displaying the precise AP pattern with the repetitive stripe pattern (Fig. 6C; Supplementary Fig. 12A). However, the analysis using submatrix B failed to reconstruct the pattern (Fig. 6D; Supplementary Fig. 12B). These results indicate that the snRNA-seq count data of the 213 genes selected at stage 7 provided essential and sufficient information for mathematically reconstructing the AP pattern. Similarly, analysis of the late stage 5 dataset showed that the 264 selected genes contained rich information for reconstructing the AP pattern (Supplementary Fig. 11D).

To investigate whether the structure of the recovered AP axis is dependent on the global or local contribution of each cluster, we modified the stage-7_nucleus full matrix by randomly exchanging gene expression values among cells of a certain cluster and performed Seurat analysis. When the 213 genes were subjected to random exchange, the ectoderm alignment became discontinuous, specifically at the modified cluster, with the nuclei

closely gathered (Fig. 6E, F; Supplementary Fig. 13A). This reconstruction failure did not occur when using other 213 genes for random exchange (Fig. 6G; Supplementary Fig. 13B). We further changed the levels of the exchange. This experiment showed that the degree of discontinuity was dependent on the percentage of nuclei in a cluster subjected to exchange (Supplementary Fig. 13C). These results indicate that each small region possesses polarity, which contributes to the generation of the global AP pattern in the entire ectoderm at stage 7, and that the expression of the 213 genes mediates the mathematical reconstruction of the AP pattern.

**Characterization of the AP axis using the changes in the expression profile**
To quantitatively capture the features of the reconstructed AP pattern, we obtained the first-derivative values of the expression profiles of the 264 selected genes for late stage 5 (Supplementary Fig. 11C) and 213 genes for stage 7 (Fig. 6B) as a function of position (from 1 to 80) along the reconstructed AP axis (Supplementary Data 12, 13). The values of the first derivative of a graph correspond to the rate of change, which directly dictates the form of the graph. We postulated that the derivative values being zero and being local maxima/minima reflected the peaks of gene expression and the boundaries of the gene expression domains, respectively. At each AP position, the derivative values were sorted, and the top-50 positive and negative values were visualized using a heatmap (Fig. 7A, B). This revealed specific regions in which many genes exhibited sharp expression boundaries. There were two such regions in the late stage-5 heatmap (Fig. 7A), one near the anterior end within the anterior terminal cluster 7, including presumptive pMES cells, and the other in the posterior terminal cluster 10, corresponding to the region where the stripe formation for the posterior segments started[62]. In contrast, the stage-7 heatmap showed a wider and more complex distribution of regions, where many genes exhibited sharp expression boundaries (Fig. 7B), indicating the progression of AP pattern development.

Next, using the first-derivative values of the gene expression profiles, we calculated the correlation coefficients between the 80 subdivisions of the

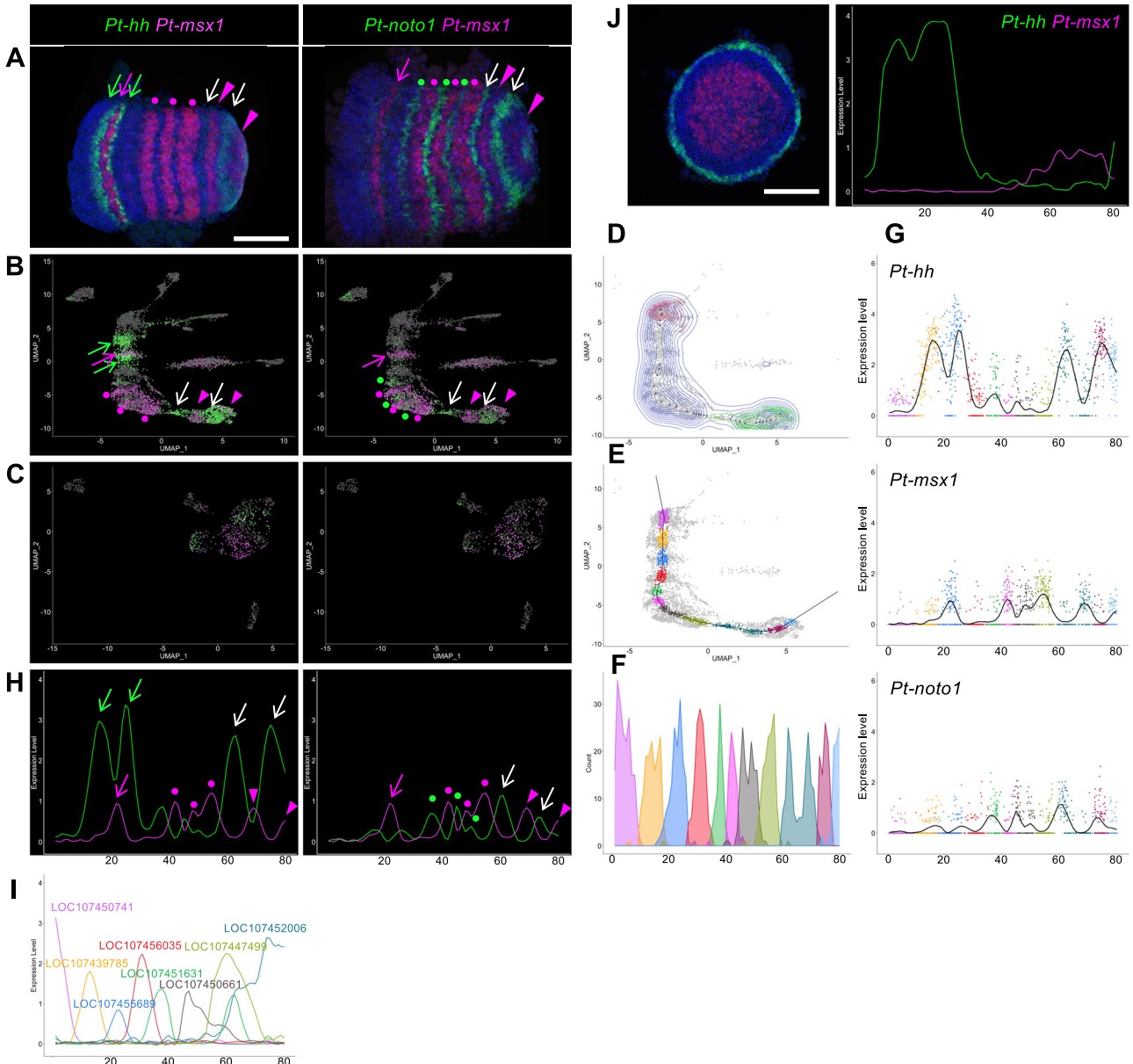

**Fig. 5 | Detection of stripe pattern and generation of expression profile.**
**A–C** Expression of *Pt-hh* (left, green), *Pt-noto1* (right, green), and *Pt-msx1* (**A** red; **B**, **C** magenta) in the stage-7 embryos (**A**) and in the stage-7_nucleus (**B**) and stage-7_cell (**C**) UMAP plots. *Pt-hh* (green arrows) and *Pt-msx1* (magenta arrows) stripes in the head, *Pt-msx1* (magenta dots) and *Pt-noto1* stripes (green dots) in the thorax, and *Pt-hh* and *Pt-noto1* stripes in L4 and O1 segments (white arrows) and *Pt-msx*1 stripes in O1 and O2 segments (magenta arrowheads) are indicated (**A**). Domains corresponding to these stripes are marked similarly (**B**). **D–G** Process for the generation of the expression profile. Contours based on the densities of ectoderm nuclei (blue) and of those expressing *Pt-six3-1* (red) and *Pt-prd2* (green), and points

plotted along the ridge (**D**). A formed spline curve (**E**) and distribution of nuclei along the linear axis (**F**). Plots of expression levels of *Pt-hh*, *Pt-msx1*, and *Pt-noto1* in the colored nuclei against the position along the axial line, overlaid with the calculated smoothing line (**G**). The color code in (**E–G**) corresponds to the cluster colors (Fig. 2A). **H** Expression profiles of *Pt-hh* and *Pt-msx1* (left) and *Pt-noto1* and *Pt-msx1* (right). The colors, arrows, and dots are the same as in (**A**). **I** Expression profiles of AP marker genes (Fig. 3A). **J** Expression of *Pt-hh* (green) and *Pt-msx1* (red) in a late stage-5 embryo (left) and the generated expression profile of these genes (right). Embryos (**A**, **J**) were counterstained with DAPI (blue), and scale bars = 200 μm.

reconstructed AP axis to determine the correlated subdivisions, where a significant number of genes showed similar changes in expression along the AP axis. The resulting 2-D heatmap for the late stage-5 data showed an ambiguous pattern; in contrast, the 2-D heatmap for the stage-7 data exhibited a repetitive pattern of positively and negatively correlated regions (Fig. 7C–F). This repetitive nature was conspicuous in the middle and posterior parts of the reconstructed AP axis, matching the forming segmental units.

We further performed the same analysis using the stage-7 data after separating the 213 selected genes into two mutually exclusive subsets: subset a, composed of 105 genes that were also included in the 264 late stage-5

genes, and subset b, composed of the other 108 genes. The patterns obtained with these two subsets were clearly different: subset a yielded a repetitive pattern in the middle part of the reconstructed AP axis (Fig. 7G), and subset b in the posterior part (Fig. 7H). These results indicate that the genes expressed from before late stage 5 and those expressed after late stage 5 differentially served as repetitive pattern elements in the middle and posterior parts of the embryo.

### Clustering of genes based on their local expression profiles
To identify genes with similar expression profiles within specific regions of the reconstructed stage-7 AP axis, we used the first-derivative values to

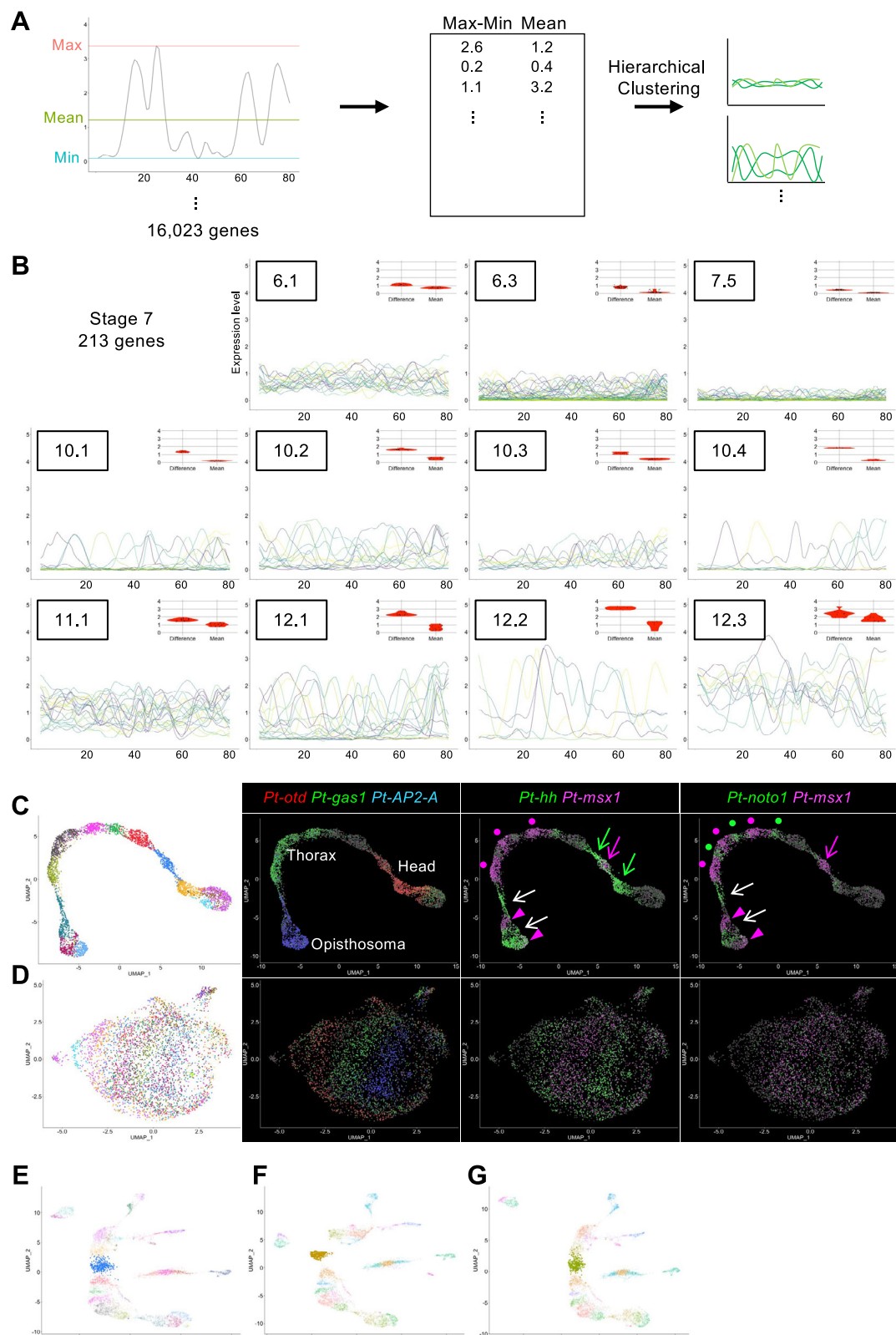

**Fig. 6 | Genome-wide search for genes with specific expression patterns.**
**A** Illustration of the search protocol. **B** Expression profiles of the 213 selected genes are visualized separately for each subgroup. Insets show violin plots of the difference and mean values of expression profiles of genes. **C, D** UMAP plots generated with submatrices A (**C**) and B (**D**). Expression patterns of body region markers and those of *Pt-hh*, *Pt-msx1*, and *Pt-noto1* are also shown. Arrows and dots point to the expression corresponding to the stripes in Fig. 5A. **E–G** Randomization of the expression of the 213 selected genes in cluster 5. Single-nucleus UMAP plots showing nuclei of cluster 5 without randomization (**E**, blue) and with randomization of the 213 selected genes (**F**, brown) and of the other 213 genes (**G**, green).

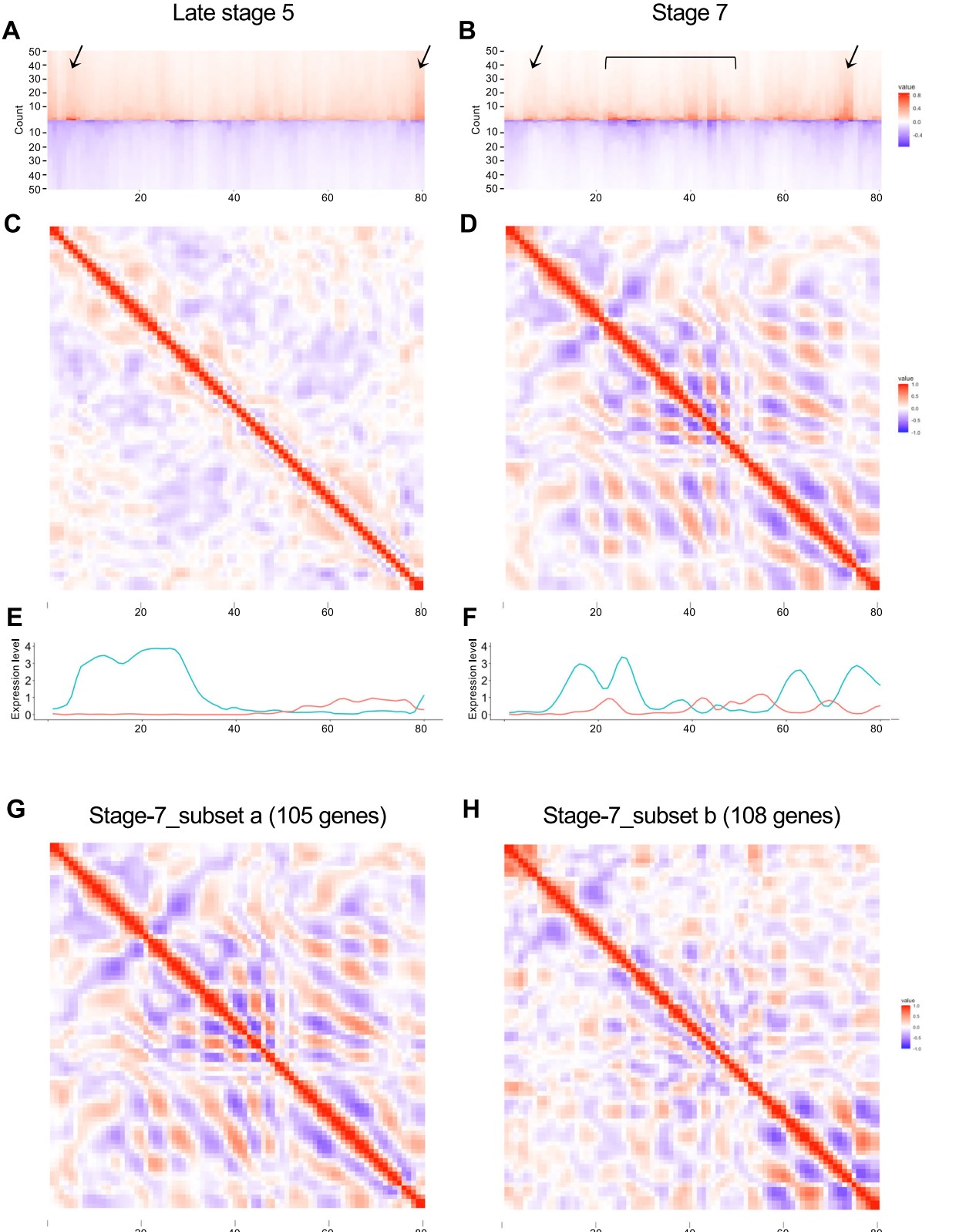

**Fig. 7 | Characterization of regions using first-derivative values. A–F** The first-derivative values of the 264 selected genes at late stage 5 and of 213 genes at stage 7 were sorted at each position, and the top 50 highest and lowest values are shown in red and blue, respectively (**A, B**). Arrows and parentheses indicate regions where high and low values are remarkable. 2-D plots of the correlation coefficients between the 80 positions, calculated using the first-derivative values of the selected 264 and 213 genes (**C, D**). Expression profiles of *Pt-hh* (blue) and *Pt-msx1* (red) at late stage 5 (**E**) and stage 7 (**F**). **G, H** 2-D plots of the correlation coefficients between the 80 positions, calculated using the stage-7 first-derivative values of 105 (subset a) and 108 (subset b) genes.

**Fig. 8 | Grouping of genes in the posterior oscillatory region using first-derivative values.**
**A** Region C at positions 55–80 in the 2-D plot of the correlation coefficients. **B** Expression profiles of the 46 genes in the six groups. **C** Heatmap showing the expression levels of the 46 genes in the single nuclei in this region. Only the colored nuclei in Fig. 5E are shown. **D** 2-D plot of the correlation coefficients between the nuclei shown in (**C**), calculated using the expression levels of the 46 genes. The color bars at the top (**C**, **D**) show the cluster colors in Fig. 2. **E**, **F** Gene pairs with high correlation coefficients. Gene pairs with $r > 0.5$ and $r < -0.5$ are connected by red and blue lines, respectively. Correlation coefficients were calculated using the expression levels in the nuclei at positions 59–72 (**E**) and 72–80 (**F**).

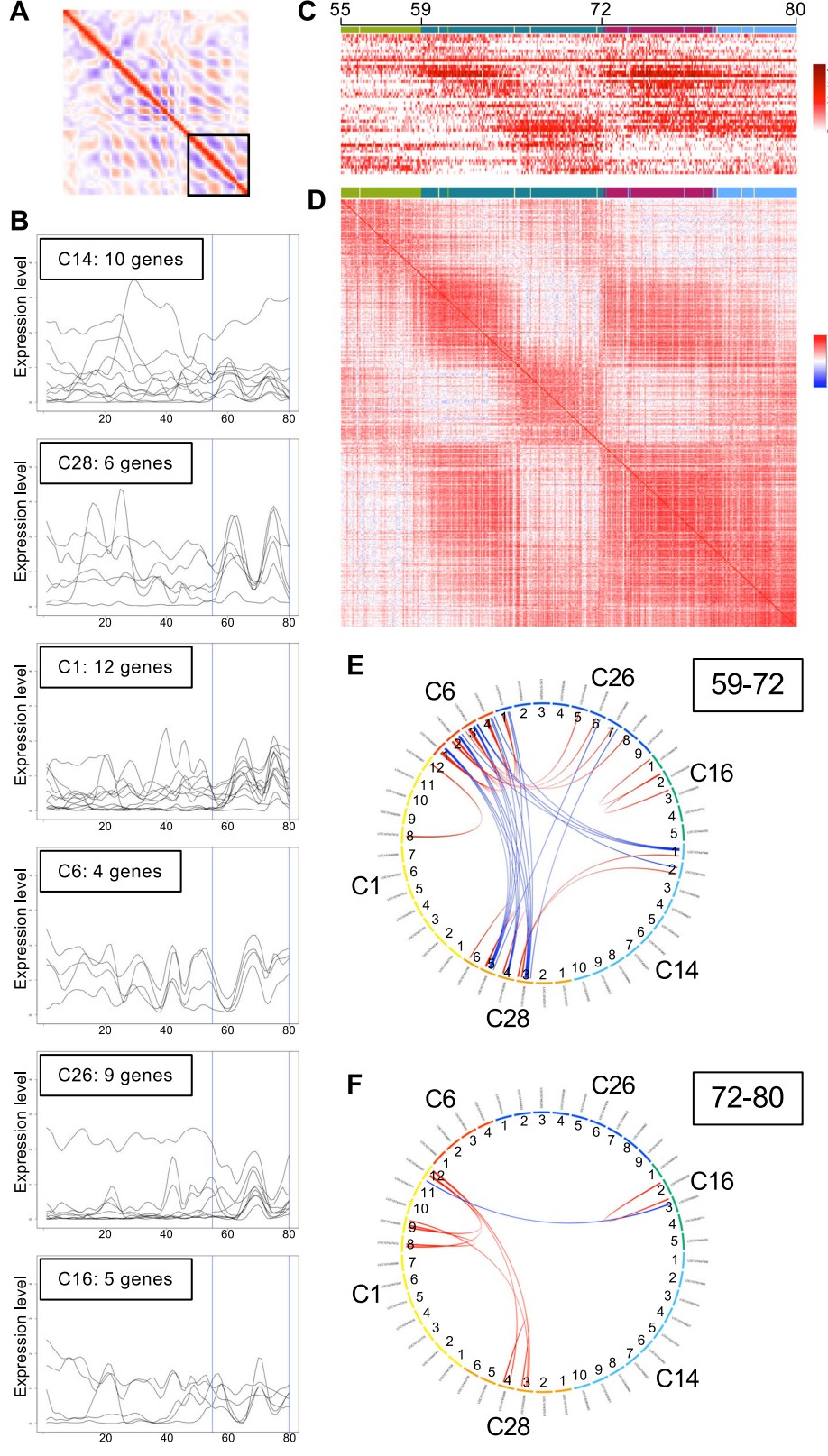

group the 213 selected genes. We set three overlapping regions, positions 1–33 (region A), 28–58 (region B), and 55–80 (region C), which approximately corresponded to the head, thorax, and opisthosoma in the stage-7 embryos, respectively. For each region, the correlation coefficients were calculated between every pair of genes using the first-derivative values, which were subjected to hierarchical clustering analysis to generate

27, 33, and 29 groups of genes for regions A, B, and C, respectively (Supplementary Figs. 14–16, Supplementary Data 14). Many of these groups contained genes with clearly recognizable and similar peak patterns, some of which showed segmental periodicity intervals. For example, group 23 from the analysis of region A (abbreviated as group A23) contained *Pt-hh* and several other genes with similar bi-splitting peaks, and group A18

contained *Pt-msx1* and several other genes exhibiting similar peak positions between the *Pt-hh* bi-splitting peaks (Supplementary Fig. 14). Groups B3, B5, B10, and B29 contained genes with three or four peaks in region B (Supplementary Fig. 15). The peak positions were similar within each group but slightly different between the groups. Similarly, in region C, groups C1, C6, C14, C16, C26, and C28 contained genes with single or double peaks at distinct positions (Fig. 8A, B; Supplementary Fig. 16), which are candidates for the clock-like oscillatory genes. Among the groups mentioned here, we noted that some shared two or three genes; groups A23 and B29 shared LOC107444253 (*Hes-1-A*), LOC107447988 (*Pt-noto1*), and LOC107451809 (*Pt-hh*); groups A23 and C14 shared LOC107447988 (*Pt-noto1*) and LOC107451464 (*Pt-hairy*); groups B3 and C1 shared LOC107456088 (*Pt-mirr4*), LOC107457313 (*Pt-sox2A*), and LOC107451770; groups B5 and C6 shared LOC107436691, LOC107439303 (*hemicentin-2*), and LOC107453597 (*Pt-fgf8*); groups B5 and C26 shared LOC107444558 (*pax-5*) and LOC107447678 (*Pt-msx1*); groups B29 and C28 shared LOC107451809 (*Pt-hh*), LOC107452888 (*Pt-winged eye-B*), and LOC107455759 (*meteorin-like*). These detected shares of genes may provide clues for investigating the whole mechanisms underlying regional variations in pattern formation. These results demonstrate the utility of the first-derivative values of the expression profiles for classifying genes according to their expression profile patterns.

### Analysis of cell states associated with posterior stripe formation using the predicted clock-like genes

Oscillatory gene expression sequentially generates new segmental units in the posterior region of the *Parasteatoda* embryo, with each unit forming within approximately 5 h[25]. The posterior unit is less developed than the anterior unit. The expression profiles of *Pt-hh*, *Pt-msx1*, and *Pt-noto1* (Fig. 5) and positional correlation analyses (Fig. 7) indicated the presence of two segmental units within region C (positions 55–80) of the reconstructed stage-7 AP axis. Gene clustering analysis (Supplementary Fig. 16) identified six groups containing 46 genes with single or double peaks in this region (Fig. 8B). Using these 46 genes, we analyzed the individual cell states in the posterior region to uncover gene-regulatory interactions. Herein, genes are referred to as, for example, 'C14 genes' for genes in group C14 and 'C14-1 gene' for the first gene in group C14.

First, a heatmap was generated, which showed the expression levels of the 46 genes in individual nuclei in region C, which were aligned according to their positions along the reconstructed axis (Fig. 8C). This heatmap revealed heterogeneity even among the nuclei at similar positions. Nonetheless, we observed two repeated units in the aligned nuclei. The repeated units were more clearly detected in a 2-D heatmap of the correlation coefficients between the nuclei, based on the expression levels of the 46 genes (Fig. 8D); the anterior unit was approximately at positions 59–72, and the posterior unit was approximately at positions 72–80. Next, within each unit, the correlation coefficients between the 46 genes were calculated using the expression levels in the single nuclei, and gene pairs with positive and negative correlations ($r > 0.5$ and $r < -0.5$) were visualized (Fig. 8E, F). While the anterior unit (positions 59–72) exhibited negative correlations between the C6 and C28 genes and positive correlations within the C6 genes (Fig. 8E), these correlations, and most other correlations detected in the anterior unit, were not detected in the posterior unit (positions 72–80) (Fig. 8F). Instead, the posterior unit showed only one negative correlation between C1-12 and C16-3 and positive correlations among C1-8, C1-9, C1-12, C28-3, and C28-4 genes. These findings indicate that dynamic changes in gene-regulatory interactions occur during stripe formation in the posterior region. Notably, the C16-3 and C1-12 genes, corresponding to *Pt-Delta* and *Pt-eve*, oscillate in different phases in the posterior terminal region of the spider embryo[19,36–38].

More comprehensive gene-to-gene correlation analyses using all the 213 genes across various regions of the reconstructed stage-7 AP axis (Supplementary Fig. 17, Supplementary Data 15) revealed distinct correlation patterns in different regions, suggesting region-specific genetic regulation during AP axis patterning in *Parasteatoda* embryos.

## Discussion

This study demonstrated that snRNA-seq data derived from stage-7 *Parasteatoda* embryos successfully reconstructed the alignment of cells along the AP axis in a UMAP plot. A faithful recapitulation of the stripe expression patterns was detected in this aligned representation. Reconstruction of an axial pattern using only data derived from scRNA-seq or snRNA-seq has not been reported in other animals. Our key observations indicate a polarized state of the ectoderm cells. Smaller clusters were aligned along the reconstructed AP axis with higher resolution parameters in the Seurat analysis (Fig. 1I). The expression of the marker genes of each ectoderm cluster was also detected in neighboring clusters at lower levels (Fig. 3G). These results emphasize the gradual transition in gene expression along this axis. Randomized gene expression within the selected ectoderm cluster disrupted the axial array at the positions of the randomized clusters (Fig. 6F; Supplementary Fig. 13A). Collectively, these observations indicate that ectoderm cells are organized along the AP axis with distinct gene expression, establishing a polarized state that underlies the formation of the stripe pattern.

The successful reconstruction of AP polarity and recapitulation of the stripe pattern in the UMAP highlight a unique aspect of the *Parasteatoda* embryo. In contrast, applying scRNA-seq and snRNA-seq to *Drosophila* embryos effectively revealed differentiating cell states, such as mesoderm, endoderm, and nervous system cells[54–56]. However, cell states corresponding to the ectoderm stripes or (para)segments are detected only in the anterior and posterior regions, and ectoderm cells in the trunk region are primarily separated into odd- and even-numbered segments[55]. Reconstructing the *Drosophila* ectoderm pattern from single-cell data requires integration of information from stained embryos[55,58,59]. This difference in the detection of stripe patterns stems from several potential factors. The use of snRNA-seq in *Parasteatoda* and scRNA-seq in *Drosophila*[55,56] may have contributed to the observed differences. Sample collection methods may also have affected the results; *Drosophila* embryos were collected in 1.5- to 4-h time windows covering various stages[54,56], while *Parasteatoda* sibling embryos were collected at essentially the same developmental stage. Furthermore, *Drosophila* embryogenesis involves simultaneous AP and dorsal–ventral (DV) patterning, leading to complex cell-state specifications, whereas *Parasteatoda* exhibits a preceding progression of AP patterning relative to DV patterning, simplifying the cell-state analysis along the AP axis[68]. These underscore the potential of the *Parasteatoda* embryo as an alternative model for dissecting the AP patterning.

Our analysis of stage-7 single-cell data revealed ectoderm superclusters corresponding to the head, thorax, and opisthosoma regions but failed to capture the precise polarity or stripe pattern. One major drawback of our scRNA-seq procedure, when using stage-7 spider embryos as the source, came from the low recovery rates of cells during sample preparation and library construction, although cell recovery was comparable between scRNA-seq and snRNA-seq when using late stage-5 embryos[62]. In general, snRNA-seq may detect fewer genes than scRNA-seq if the scale of the data is comparable[69]. Our snRNA-seq procedure, however, can have remarkably high recovery rates of nuclei from a small number of spider embryos, even at stage 7 (Table 1), while detecting more genes than scRNA-seq as a whole. The differences between scRNA-seq and snRNA-seq may have other aspects. The discrepancy in detecting the complex stripe pattern may reflect differences in sensitivity to temporal changes in transcription between the two techniques. Nascent pre-mRNAs captured in single-nucleus data may provide a more dynamic view of transcriptional changes than mature mRNAs captured in single-cell data. This hypothesis is supported by the observation that single-nucleus data obtained from late stage-5 sibling embryos clearly detected the cell states emerging from the germ disc center for posterior segmentation, which was not apparent in the single-cell data[62]. However, other factors may also contribute to these differences, including variations in the number of cells and nuclei, sample preparation procedures, and slight differences in developmental stage between the embryos used for the library construction.

Our analysis revealed several novel cell states. At least two endoderm cell states were detected at late stage 5 (Fig. 4A, B; Supplementary Fig. 4),

suggesting heterogeneity within the endoderm cell population beyond the previously described pEND and cEND cells[36]. These differences likely reflect the degree of differentiation. Despite originating from different regions (the center and periphery of the germ disc), stage-7 endoderm cells exhibited a relatively uniform cell state (Fig. 4C, D; Supplementary Fig. 8). In contrast, four distinct mesoderm clusters were identified at stage 7, reflecting differences along the AP axis. Further investigation is needed to determine whether mesoderm cells exhibit further AP polarization in later stages and to understand the potential interactions between the mesoderm and overlying ectoderm in shaping their respective cell fates. The classification of CM cells has been a subject of debate[62,70,71]. Our data showed a close proximity between CM cells and endoderm cells (Fig. 1C), indicating that CM cells may represent a specialized subpopulation of the endoderm.

We established a novel method to generate quantitative gene expression profiles along the reconstructed AP axis in a UMAP plot (Fig. 5; Supplementary Fig. 9). These profiles accurately reflected the gene expression patterns along the AP axis in the *Parasteatoda* embryo, including distinct phases of the segmental stripes. This approach enabled genome-wide, unbiased identification of genes exhibiting specific AP expression patterns. We identified 213 genes from stage-7_nucleus data and 264 genes from late stage-5_nucleus data that could be used to reconstruct the AP pattern, including the segmental stripe pattern at stage 7 (Fig. 6; Supplementary Fig. 11). Although this does not definitively prove that these genes directly establish the AP pattern during embryogenesis, it provides crucial clues for identifying the underlying molecular networks. By analyzing the first derivatives of the expression profiles, we identified gene groups with similar peak patterns (Supplementary Figs. 14–16), which are candidate genes involved in stripe formation. Furthermore, we examined gene expression in the posterior region at single-cell resolution to identify gene pairs that tend to be co-expressed or expressed in a mutually exclusive manner (Fig. 8E, F, Supplementary Fig. 17). This analysis provides valuable data for investigating potential gene interactions. These analytical approaches can be applied to embryos at different developmental stages. As the *Parasteatoda* spider lays hundreds of eggs simultaneously, we can analyze serial pools of embryos collected according to time-based staging. Embryos subjected to gene knockdown experiments can be analyzed similarly. These advantages of using spider embryos facilitate a further investigation of the dynamic cell-state regulation of the stripe-forming processes. The high-resolution, quantitative single-nucleus data generated in this study provide a robust foundation for a deeper understanding of the diverse mechanisms underlying the arthropod segmented body axis and offer a reliable connection to the mathematical modeling of cell-based patterning processes.

This study demonstrates that quantitative data representing key characters of an animal phylum can be obtained based on the genome of an organism without relying on information from established model organisms. This finding opens new avenues for exploring the origins of animal diversity.

## Methods
### Spider
Animal experiments were performed according to a protocol reviewed and approved by the Institutional Animal Care and Use Committee of JT Biohistory Research Hall (2020-5). We used laboratory stocks of the spider *Parasteatoda tepidariorum* (synonym, *Achaearanea tepidariorum*), maintained at 25 °C under 16-h light and 8-h dark photoperiods. The developmental stages of the embryos have been described previously[72]. Stage-7 embryos were collected 13 h after late stage 5, when the cumulus reached the rim of the germ disc. For single-cell and single-nucleus isolation and embryo fixation, eggs derived from one egg sac were used for each experiment; 20–120 eggs were dechorionated, and approximately half of the embryos that displayed a typical morphological appearance were selected. To ensure the reliability of the data, the development of 8–16 sibling embryos was carefully monitored in each experiment to confirm their normal development.

### Isolation of the cell and the nucleus
Spider embryos were dechorionated using commercial bleach for 4 min, and then rinsed with distilled water. Embryos were then transferred into CGBS-CMF [Chan and Gehring buffered saline[73], Ca- and Mg-free: 55 mM NaCl, 40 mM KCl, 10 mM Tricine, pH 6.95] solution containing 0.01% (for cell) or 0.5% (for nucleus) bovine serum albumin (BSA: Nacalai Tesque, 01278-44). The vitelline membrane was manually removed with forceps. For single-cell isolation, devitellinized embryos were incubated in CGBS-CMF containing 1 mg/mL elastase solution (Worthington) at room temperature for 5 min. The cell suspension was then filtered through a 70-μm strainer (pluriStrainer-Mini, 43-10070) followed by two passes through a 10-μm strainer (pluriStrainer-Mini, 43-10010) with brief centrifugation. Cells were collected in a DNA LoBind tube (Eppendorf, 022431021). The elastase reaction was terminated by adding BSA to a final concentration of 1%.

Single-nucleus isolation was performed as previously described[62]. In brief, devitellinized embryos were resuspended in 400 μL homogenization buffer [250 mM sucrose, 10 mM Tris (pH 8.0), 25 mM KCl, 5 mM MgCl$_2$, 0.1% Triton-X100, 0.1 mM DTT, 100 U/mL RNase inhibitor (SUPERaseIn RNase Inhibitor: Invitrogen, AM2694), and protease inhibitor at 1:100 dilution (Nacalai Tesque, 25955-11)][74] and homogenized using a loose plastic pestle (Fisher scientific, 12-141-368) with 40 strokes. The homogenate was centrifuged, and the precipitate was resuspended in CGBS-CMF containing 0.5% BSA and 100 U/mL RNase inhibitor. The suspension was filtered through a 40-μm strainer (pluriStrainer-Mini, 43-10040) and washed with the same solution. Isolated single-cell and single-nucleus samples were stored with five volumes of CellCover (Anacyte Laboratories, 800-050) at 4 °C until further use.

Fifteen and twenty embryos from different egg sacs were used for the single-cell and single-nucleus isolation, respectively. Concurrent with the cell and nucleus isolation, sibling embryos were fixed for subsequent staining.

### Construction and sequencing of libraries
scRNA-seq and snRNA-seq libraries were constructed using a BD Rhapsody WTA Amplification Kit (BD Biosciences, 665915) following the manufacturer's protocol with modifications. For scRNA-seq construction, the stored sample was centrifuged, resuspended in a 0.5× CGBS-CMF containing 0.01% BSA, and loaded onto a cartridge pre-rinsed with 0.5× CGBS-CMF. For snRNA-seq library construction, RNase inhibitor was added to the sample buffer at a final concentration of 100 U/mL, Proteinase K was added to the lysis buffer at a final concentration of 0.5 mg/mL, and the lysis step was extended to 5 min. Sequencing was performed using the Illumina HiSeqX Sequencing System with 150-bp paired-end reads. We obtained 435,917,554 and 485,625,106 raw reads for scRNA-seq and snRNA-seq, respectively. The data were deposited in the GEO repository (GSE287446).

### Generation of gene expression matrices
RNA-seq reads were processed using the BD Rhapsody WTA pipeline on the Seven Bridges Genomics Cloud platform[75] with default parameters, as described previously[62]. Low-quality reads were then filtered. Cells/nuclei and RNA molecules were identified based on the cell barcode and unique molecular index (UMI) information contained within the R1 reads. The R2 reads were aligned against the *Parasteatoda tepidariorum* genome[45], GCF_000365465.3_Ptep_3.0, using the STAR index created by STAR-2.5.2b[76]. For snRNA-seq data, the annotation file accompanying the genome information was modified to include both exonic and intronic reads for read counting. The alignment results were integrated with the cell barcode and UMI information to generate an expression matrix for each sample. These expression matrices were deposited in the GEO repository associated with the RNA-seq data.

### Clustering analysis with Seurat
Data were analyzed using the Seurat package (v.4.1.1)[77,78]. To filter out low-quality cells and nuclei and potential doublets, we selected cells and nuclei

based on the number of unique genes and total RNA molecules. Cells with more than 1500 unique genes and fewer than 20,000 total RNA molecules were selected for stage-7_cell data. Similarly, nuclei with more than 2000 unique genes and fewer than 20,000 total RNA molecules were selected for stage-7_nucleus data. The late stage-5_nucleus data[62] were reprocessed, selecting nuclei with more than 1500 unique genes and fewer than 15,000 total RNA molecules (Supplementary Fig. 2). During the filtering, we removed 681 nuclei from the late stage-5_nucleus data, 152 cells from the stage-7_cell data, and 2182 nuclei from the stage-7_nucleus data. This resulted in the final cell/nuclei counts presented in Table 1.

Following the instructions provided with the Seurat package, we performed log-normalization, identified 2000 variable features, scaled the data, and conducted principal component (PC) analysis to reduce the dimensionality of the data to 50. The processed data were then subjected to FindNeighbors and FindClusters functions, followed by nonlinear dimension reduction using UMAP (Uniform manifold approximation and projection). The data were visualized in a 2-D space (Fig. 1C–E). Multiple parameter settings for the dimension and resolution were tested (Fig. 1I; Supplementary Figs. 3, 5). Additionally, we performed Seurat analyses on the pre-filtered dataset and observed that the excluded cells/nuclei predominantly localized to clusters exhibiting lower counts in the UMAP (Supplementary Fig. 2). This observation suggests that the filtering step had minimal effects, if any, on the downstream analyses.

### Random selection of nuclei from the expression matrix
To analyze the stage-7_nucleus data with a sample size equivalent to that of the stage-7_cell data (865 nuclei), 865 nuclei were randomly selected from the full stage-7_nucleus matrix using the sample function in R. A submatrix was generated using these selected nuclei. Subsequently, Seurat clustering analysis was performed on this submatrix with the following parameters: dimension, 1:50; resolution, 2.5. This random selection and subsequent analysis were independently repeated 10 times (Supplementary Fig. 6).

### Identification of cluster marker genes
The marker genes for each cluster were identified using the FindMarkers function in Seurat. Genes expressed in more than 25% of nuclei in a cluster and less than 90% in other clusters with adjusted $p$-values less than 0.001 were listed as markers for the cluster (Supplementary Data 2, 3).

### Visualization of gene expression on the UMAP plot
Gene expression was visualized on the UMAP plot using the FeaturePlot function of the Seurat package. To visualize the expression of two or three genes simultaneously, feature plots were generated using a gray75 to black color scale. These plots were then converted into negative images, overlaid, and color-coded using RGB values in ImageJ (Fiji) software (ver. 2.9.0/1.53t). UMAP plots depicting the expression of all genes in the expression matrices can be accessed through our web database and have been deposited in Figshare[66,67].

### Assignment of nucleus positions along the reconstructed axis
In the UMAP plot of the stage-7_nucleus RNA-seq data generated with the clustering parameters, a dimension of 1:50 and a resolution of 0.5 (Fig. 1D), a contour encompassing ectoderm clusters 0, 1, 2, 4, and 5 was drawn based on the density of 4129 nuclei using the geom_density_2d function in the ggplot2 package (v.3.4.2). Contours were also drawn for nuclei expressing Pt-six3-1 and Pt-prd2; the former was expressed at the most anterior part of the germ band, and the latter was expressed in oscillatory stripes in the posterior region (Supplementary Fig. 9A). These were used to determine the anterior and posterior ends. Using Python (v.3.8.3), several points were manually plotted along the ectoderm contour ridge and at the Pt-six3-1 and Pt-prd2 contour peaks. Two additional points were plotted on the extension lines from the anterior and posterior termini. These extensions were used to correctly calculate the distances from the nuclei to the reconstructed axis around the anterior and posterior ends, as described below. A spline curve

was fitted through these points using the xspline function of R (v.4.2.1), which represents the reconstructed AP axis (Fig. 5E).

To assign the position of plotted nuclei, the axial spline curve was evenly divided into 10,000 points using Numpy (v.1.22.4). The positions of these points along the curve were determined by summing the distances between the neighboring points from the anterior end. The nearest point on the curve was identified for each nucleus in the UMAP plot. The position of each nucleus was assigned as the position of its nearest point on the curve and the distance to the nearest point. Nuclei located more anterior or posterior than the ends of the spline curve were omitted.

A similar analysis was performed on the UMAP plot of the late stage-5_nucleus data (dimension: 1:45; resolution: 0.5) (Supplementary Fig. 9D) using 4230 nuclei from ectoderm, mesoderm, and extraembryonic clusters 0, 1, 2, 3, 5, and 6. The anterior and posterior ends were defined based on the expression of Pt-twist and Pt-aslH. Nuclei located near the anterior end and not expressing Pt-twist were omitted. The positions of the nuclei are listed in Supplementary Data 4.1 and 6.1.

### Generation of gene expression profiles
Nuclei within 0.3 units of the axial curve were selected for generating quantitative gene expression profiles (shown in Fig. 5E and Supplementary Fig. 9D, colored nuclei), specifically 1387 nuclei at stage 7 and 1138 nuclei at late stage 5 (Supplementary Data 4.2, 6.2). To mitigate the potential impact of the limited number of nuclei near the anterior and posterior ends on the subsequent smoothing step, the data were mirrored: nuclei from the anterior half were mirrored at the anterior end, and those from the posterior half were mirrored at the posterior end, resulting in a doubled expression matrix. Loess smoothing was performed with the geom_smooth function in ggplot2 (span = 0.05) using the nuclear positions in the doubled matrix and their normalized gene expression counts. This generated a smoothed line with 160 points for the expression profile of each gene. The mirrored data were then reduced to 80 points by selecting points 41–120 for downstream analyses. Gene expression profiles at late stage 5 and stage 7 are shown in Supplementary Data 7 and 5. The patterns of the expression profiles are available in our web database and the Figshare repository[66,67].

### Search for genes with specific expression patterns along the AP axis
First, to exclude genes expressed in small numbers of sparsely distributed cells, which could be difficult to distinguish from genes expressed in a region-specific manner, we determined the minimum distance between nuclei expressing each gene along the reconstructed AP axis. Based on these minimum distances, Euclidean distances were calculated between genes, and hierarchical clustering was performed using ward.D2 method, resulting in two gene groups. A small group of 946 sparsely expressed genes was excluded from further analysis (Supplementary Data 16), and a larger group of 16,023 genes was retained for subsequent analyses.

For each of the 16,023 genes, two values were extracted from their expression profiles: the mean expression level and the difference between the maximum and minimum expression (Supplementary Data 8). The Euclidean distances between genes were calculated based on these two values, and hierarchical clustering was performed. Several groups were selected based on relatively low mean expression levels and large differences in expression. Subsequently, genes within each selected group were subjected to a second round of clustering analysis. This process resulted in the selection of 213 genes at stage 7 (Supplementary Data 9). Additionally, we conducted the second-round clustering for other unselected groups, which resulted in forming subgroups that did not match the selection criteria (Supplementary Data 8). A similar analysis was performed on the late stage-5 data. After excluding 1149 sparsely expressed genes (Supplementary Data 17), the remaining 15,061 genes were subjected to two rounds of clustering to select 264 genes (Supplementary Data 10, 11).

## Construction of UMAP with the selected genes

The normalized stage-7_nucleus expression matrix containing 19,692 genes and 6239 nuclei was reduced to two smaller matrices: matrix A, containing the 213 selected genes and 4102 ectoderm nuclei, and matrix B, containing the remaining 19,479 genes and the same 4102 ectoderm nuclei. Similarly, two matrices were generated from the late stage-5_nucleus data: one containing 264 selected genes and 4230 nuclei, and the other containing 18,835 remaining genes and 4230 nuclei. These reduced matrices were then used for subsequent Seurat analyses (Fig. 6C, D; Supplementary Figs. 11D, 12). To visualize the expression of the excluded genes (the 213 and 264 genes, respectively) in the UMAP, their expression values were extracted from the original (unreduced) matrix using the IDs of the corresponding nuclei.

## Randomization of the expression of genes in a cluster

To randomize the expression of the 213 genes within the selected ectoderm clusters in the stage-7_nucleus count matrix, we followed the procedure described below. A set of random numbers, equal to the number of nuclei in the cluster, was generated using the sample function in R. This process was repeated to create 213 sets of random numbers corresponding to the 213 genes. Based on each set of random numbers, the expression values of a specific gene were randomly exchanged among nuclei within the selected cluster. In the first experiment, randomization was performed for each ectoderm cluster using the selected 213 genes (Fig. 6E, F; Supplementary Fig. 13A). In the second experiment, randomization was performed for cluster 5 using the other 213 genes randomly selected from among the highly variable genes identified in the Seurat analysis. This experiment was repeated three times (Fig. 6G; Supplementary Fig. 13B). In the third experiment, randomization was performed for cluster 5 using the original 213 genes. However, in this experiment, the expression values were exchanged among a subset of cluster-5 nuclei (10–90% of the cluster, selected independently for each gene). Randomized matrices were used for subsequent Seurat analyses (Supplementary Fig. 13C).

## Calculation of the first-derivative value and correlation analysis

First-derivative values were calculated for each of the 80 positions in the expression profiles of the 213 stage-7 and 264 late stage-5 genes predicted to exhibit specific expression patterns (Supplementary Data 12, 13). For positions 2–79, the derivative was calculated as the average difference in expression levels between the current position and its two neighboring positions. For the terminal positions (1 and 80), the derivative was calculated as the difference between the current position and its single neighboring position (positions 2 and 79, respectively). The top 50 highest and lowest first-derivative values at each position are shown in Fig. 7A, B.

Using the first-derivative values of all these genes at 80 positions, Pearson correlation coefficients were calculated between the 80 positions using the cor function in R. These correlation coefficients were visualized in a heatmap (Fig. 7C, D). Similarly, the correlation coefficients calculated with subsets of stage-7 genes are shown in Fig. 7G, H.

## Grouping of genes using the first-derivative values

To group genes with similar expression profile patterns within the three regions (region A, positions 1–33; B, 28–58; and C, 55–80), the following approach was used: the 213 stage-7 genes were filtered to exclude those with the maximum values of the expression profiles less than 0.2 within the target region. Pearson correlation coefficients were calculated between genes based on their first-derivative values within the target region. Euclidean distances were then calculated from the correlation coefficients, and hierarchical clustering was performed using the ward.D2 method to group the genes. This resulted in 27 groups in region A, 33 groups in region B, and 29 groups in region C (Supplementary Figs. 14–16).

## Expression in single nuclei and correlation analysis

From the normalized expression matrix of the stage-7_nucleus data, we selected the colored nuclei depicted in Fig. 5E, along with the 213 genes predicted to exhibit specific patterns. Several submatrices were generated using this selected matrix. Each submatrix included nuclei located within a specific target region and genes expressed in more than 10% of the nuclei within the target region. For each submatrix, the correlation coefficients between cells and between genes were calculated using the cor function in R. Gene pairs with relatively high correlation coefficients ($r < -0.5$ and $r > 0.5$) were visualized by connecting them with blue and red lines, respectively (Fig. 8E, F; Supplementary Fig. 17C).

## cDNA cloning

Full-length or partial cDNAs were obtained from our laboratory stocks of EST clones or isolated using PCR. The cDNAs were used to synthesize probes for in situ hybridization. The cDNA clones and the PCR primers used in this study are listed in Supplementary Data 18.

## Fluorescence in situ hybridization

Embryo fixation and FISH were performed as described previously[79]. DIG-, DNP-, and FITC-labeled probes (Table 2; Supplementary Data 18)[80,81] were used and detected with anti-DIG-POD (1:1000 dilution; Roche 11 207 733 910), anti-DNP-HRP (1:200 dilution, PerkinElmer FP1129), and anti-FITC-HRP (1:200 dilution, PerkinElmer NEF710) antibodies in combination with 5-(and-6)-carboxyfluorescein and DyLight680 tyramides. Samples were counterstained with 4′,6-diamidino-2-phenylindole (DAPI; Sigma-Aldrich) to visualize DNA. The stained embryos were individually mounted on glass slides with spacers and observed using a TCS SPE confocal system (Leica). Images were processed using ImageJ (Fiji) software ver. 2.14.0/1.54 f. Z-sections were projected using Max projection.

## Statistics and reproducibility

For all statistical analyses, the methods used are addressed in the relevant sections. For sample collection, embryos with a typical morphological appearance were randomly selected. For single-sell and single-nucleus data analyses, cells/nuclei expressing genes/molecules within the range of numbers specified in the "Methods" section were retained.

## Reporting summary

Further information on research design is available in the Nature Portfolio Reporting Summary linked to this article.

## Data availability

The stage-7 scRNA-seq and snRNA-seq reads and processed data have been deposited in GEO under accession number GSE287446. The late stage-5 snRNA-seq reads and the original matrix are available at GEO under accession number GSM6069861 in the data series GSE201705[62], and the reprocessed late stage-5 data are available in GSE287446. Gene expression patterns in the UMAP and patterns of the expression profiles have been deposited in figshare[66,67] and are also available on our website (https://www.brh2.jp). Material requests should be addressed to yasuko@brh.co.jp.

## Code availability

The scripts used for analyzing the data and producing graphs and figures are available at https://github.com/Takanori-Akaiwa.

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

## Acknowledgements

We thank A. Noda for technical assistance; Prof. Y. Takahashi, Prof. K. Sano, and Prof. T. Nakano for encouragement and discussion; and the members of JT Biohistory Research Hall and the members of the Multicellular System Area, PRESTO JST, for discussion. This work was supported by JST PRESTO (JPMJPR2041) to Y.A. and JSPS Kakenhi (20K06676, 24K01771) to Y.A., and the JT Biohistory Research Hall.

## Author contributions

Y.A. and H.O. conceived the project, which was registered as a research plan of the PRESTO and BRH. T.A. conducted the laboratory experiments with help from Y.A. and H.O., and T.A. and Y.A. analyzed the data with support from H.O. Y.A. wrote the manuscript with support from H.O.

## Competing interests

The authors declare no competing interests.
