## [Transparent Peer Review file · Communications Biology]

Genome-wide quantitative dissection of an arthropod segmented body plan at single-cell resolution

Corresponding Author: Dr Yasuko Akiyama-Oda

This manuscript has been previously submitted to another journal. This document only contains information relating to versions considered at Communications Biology.

Version 0:

Reviewer comments:

Reviewer #1

(Remarks to the Author)

The broader context of the paper is understanding arthropod segmentation mechanisms, which are not universal across arthropods, and may contribute to understanding how a homologous process (segmentation) can be achieved in multiple molecular ways. The manuscript by Akaiwa, Oda & Oda is an in-depth single cell genomics characterization of two early embryonic stages of spider development. Though a part of the data has been published earlier for one stage (stage 5, radially symmetric disc), they sequence single-cell and single nuclei for stage 07 (antero-posterior axis patterning), and also single-nuclei for stage 5. The most striking feature of the descriptive part of the paper is the recapitulation of the A-P axis into the 2D UMAP representation of the multidimension data, which is based on expression levels of each cell/nucleus.

The paper can be divided in two components:

(1) description of the single cell data for both stages, with variation in multiple parameter, validation and cluster annotation with in situ hybridization.

(2) Method development and genome-wide expression pattern prediction.

This is a beautifully done study. Though a few single-cell datasets have been published for spiders, this study pushes the boundary in that it provides unparalleled resolution and genome-wide biological insights for the question of segmentation. The characterization of the dataset is exemplary, and I appreciate the exploration of multiple parameters for the UMAPS to show that the head/trunk/tail alignment in the UMAP is robust. The resolution of the data is impressive, and the clusters, even the ones composed of few cells, are well annotated with marker genes, at least in the single-nuclei data. They present and thoroughly characterize the ectoderm cluster, showing that specific regions along the A-P axis of the embryo are represented by nuclei along a line in the UMAP ectodermal cluster

Given that the UMAP reflects the AP axis very well (unusual), they devise a method to approximate a line along the clusters and quantify expression patterns along the line. Even though the method they developed might not work in the context of other datasets it worked very well for these early embryos, allowing for the discovery of candidate A-P genes. This method recapitulated benchmark genes, and allowed, via clustering, identifying genes with similar patterns. They find that around 200 genes are sufficient to encode the AP information, though data resampling analyses.

I strongly recommend publication: This a great dataset, a show-case of the power of genomics in lifting the rigor of research in emerging model organisms, and foundational for the question of segmentation in arthropods. I have mostly minor comments and requests for clarification.

General comment:

-Color and dot sizes in the UMAPS. The black UMAPS with small dot sizes are very difficult to visualize (ex: Fig 1D, center). I imagine the choice was made to draw a correspondence with the colors in the FISH. If the authors prefer the black background, they could consider increasing dot sizes to improve visualization, and using a lighter gray to show the shape of the clusters better. Alternatively, the UMAPS could all be shown with white background, as the colors would stand out much more clearly.

Minor comments:

Introduction

The introduction is excellent.

Line 32: I suggest changing for "may be remarkably diverse", because there are certainly cases where it is conserved.

Line 65: I think genome assembly is more accurate.

Results and Discussion

Line 130–133: Typos in nucleus

Line 134–135: Could you please briefly elaborate in the text why they may be pseudo clusters? Such as why would all clusters are expected to have roughly the same number of expressed genes.

Figure S3: It would be helpful to add the gene names under the LOC number, in addition to their mention in the figure legend.

Line 154: It would be helpful to mention in the text (in parenthesis) what the markers for each tissue are called

Lines 151–158: It took me a few passes to understand what was meant here, about the split on not-split cluster to the endo/ex cluster in 1D and 1E. Could you please clarify this section? I also did not understand how the presented data support of not the hypothesis about the origin of the extra embryonic cells. In this passage, I think it would be useful to add a sentence about what is the implication of this difference between nucleus or cell data, or at least what difference we should expect between both datasets.

Line 165: Should it be the third graph?

Line 169: In Figure 3A, some locus projections are not restricted to only the clusters pointed by the arrows. For example, LOC107451631. Here I think it is worthy at least mentioning this, because when stating "perfectly" in this sentence one wonders if expression in the other clusters is real or not.

Lines 189–191: I think this sentence is circular.

Line 312: The in silico experiments of modifying expression matrices are fascinating. My suggestion for here and the previous instances is to open the paragraph with a clearer explanation of what the goal of the experiment is, instead of "to further investigate their contribution". In this paragraph and before, it would help a lot the reader in following the experiment. So in this case, something as: "To investigate the if the structure of the recovered AP axis is dependent on a global vs local contribution of individual clusters, we..."

Line 369: trough?

Line 421: selected ectoderm cluster

Line 618: remove "that"

Additional comments:

-Are there genes that are repeated between sets A, B and C? B29 and A23 seem to have some shared genes, for example.

-It would be beneficial to explain what the first derivative is somewhere in the Results or Methods.

Reviewer #2

(Remarks to the Author)

Summary:

This manuscript presents a comprehensive single nucleus and single cell RNA-seq analysis of segmentation in the spider *Parasteatoda tepidariorum*. This is the third embryonic single-cell sequencing project for this species in the last couple of years, but it focuses on a different subset of developmental stages and asks different questions. The manuscript offers genome-wide, spatially resolved insights into anterior-posterior (AP) patterning. The authors demonstrate that ectodermal AP polarity and segmental stripe formation can be reconstructed from transcriptomic data alone, enabling a novel approach to studying developmental diversity in arthropods. This work represents a significant contribution to evolutionary developmental biology and single-cell genomics. I have no significant criticisms of the manuscript, but only a few technical comments and suggestions for improvement.

General comments:

The readers could benefit from a very brief discussion of the pros and cons of sn/sc use.

Figures presenting UMAPs with a black background are very difficult to see except at high magnification. It is not clear why a black background was chosen for these figures. If the authors wish to differentiate these panels (highlighting expression of specific genes within the UMAP), I suggest using a different colored panel outline or something similar, rather than a black background.

Using "stage 5l" is confusing due to the similarity of a lower-case l and the numeral 1 in different fonts. Indeed, the age in hours of Stage 7 is 51 hours, causing additional confusion. I suggest using "5late" or a similar annotation.

Specific comments:

Line 46-47 - the sentence refers to differences in genome size and genome content, but the cited references refer to gene content and to whole genome duplications in spiders. Discussion of genome size evolution usually refer to a more continuous change in genome size, and not specifically to duplications. I suggest omitting "genome size" in this context.

Line 73 - "bifurcating cell trajectories" - why only bifurcating?

Line 252 - "genome widely predict" sounds awkward. I suggest changing to "predict in a genome wide fashion" or something similar.

Ariel Chipman

Reviewer #3

(Remarks to the Author)

The authors used single-cell and single-nucleus RNAseq data to reconstruct the emergence of AP polarity and segmentation in the common house spider *Parasteatoda tepidariorum*. The careful data analysis reaches an impressive degree of spatial refinement yielding UMAPs with ectodermal clusters corresponding to individual segment analgen. The data set also uncovers new mesodermal and endodermal clusters. Together these data will be the foundation of future functional studies which promise substantial new insights into spider segmentation and germ layer development. The paper is well written and should be of interest to a broad readership of researchers interested in single cell approaches in general and arthropod segmentation.

Reviewer #4

(Remarks to the Author)

The authors of the manuscript titled "Genome-wide quantitative dissection of an arthropod segmented body plan at single-cell resolution" presented a single cell transcriptome study for the two developmental stages that related to AP axis segmentation emergence. The authors separately analyzed the scRNA-seq/snRNA-seq datasets related to their projects , discovered and validated many embryo body segments specific marker genes. Their studies could be valuable for the developmental study of spiders. However, I have several concerns regarding their current data analyses, which are listed as followed.

(1) The authors mentioned that the two stages chosen for their study represents a critical period of emergence of segmental units along the AP axis. Although the author presented analyses for each stage embryos separately as shown in Figure 1C-E, however, it might be better to directly do an integrated analysis among datasets, which could align the cells across developmental stages.

(2) Why the snRNA-seq used in Figure 1G captured more genes than the scRNA-seq used in Figure 1H? As my understanding, snRNA-seq technology normally capture less gene transcripts than scRNA-seq.

(3) As the authors mentioned in the method part (line 557-571), the authors only considered the cells with more than 1500 unique genes (more than 2000 genes for snRNA-seq). Could these criteria too stringent for single-cell transcriptome study? Based on previous published single-cell transcriptomic studies, the cells with unique genes between 200 and 1500 still can be useful. Could the authors provide more detail about the data before and after filtering? Such as the number of cells before and after filtering and the distribution of the number of genes before and after filtering.

Reviewer #5

(Remarks to the Author)

Version 1:

Reviewer comments:

Reviewer #1

(Remarks to the Author)

The authors have addressed all my comments in the rebuttal letter and I have no further suggestions. Congratulations on the nice work. My recommendation is to accept the manuscript.

Reviewer #2

(Remarks to the Author)

The authors have addressed all of my comments and the comments of the other reviewers. I have no further requests or suggestions.

Reviewer #4

(Remarks to the Author)

No further concerns for the revised manuscript.

Reviewer #5

(Remarks to the Author)

Manuscript title: Genome-wide quantitative dissection of an arthropod segmented body plan at single-cell resolution

Authors: T. Akaiwa, H. Oda, and Y Akiyama-Oda

We would like to thank all the reviewers for their constructive suggestions regarding on our manuscript. We have incorporated as many of these suggestions as possible into the revised version.

Sincerely yours,

Yasuko Akiyama-Oda

Points of Revision

The major points of revision are as follows:

1. In the UMAP plots, larger point size and lighter gray color were implemented.
2. Information regarding the number of genes and molecules per cell and nucleus before and after filtering is presented in the newly included Supplementary figure 2 (attached in the last page of this file). This figure also shows the results of Seurat clustering performed on the pre-filtered counts. The corresponding text describing the filtering and clustering was also incorporated (line 593–596, 603–607).
3. To facilitate the recognition of marker genes, gene names have been added alongside LOC numbers for known genes in the text and in Fig. 3, Supplementary figs 4 and 8. Furthermore, Table 2, which shows a list of cell/tissue marker genes, has been reproduced from Table S2.
4. Unclear phrases and sentences have been revised in accordance with the reviewers' suggestions (line 33, 48, 66, 74, 156–158, 194–195, 258, 318–319, 655).
5. Explanations concerning the pseudoclusters are included (line 134–138).
6. A sentence concerning the two expression domains of LOC107451631 has been included (line 173–174). In Fig. 3A, arrowheads have been added to indicate the expression domains within the stained embryos.
7. A sentence explaining the first derivative values has been added (line 335–337).
8. Information concerning shared genes among the groups has been included (line 383–392).
9. Sentences concerning the distinction between scRNA-seq and snRNA-seq have been included (line 465–472).
10. 'Stage 5l' was changed to 'late stage 5'.

11. We submitted information on the reprocessed late stage-5 data to GEO, which is available under the same accession number (GSE287446) as the stage-7 data. This information has been included (line 994).
12. Typos and grammatical errors have been corrected.
13. The format has been adapted to the journal specifications.

Point-by-point responses to the comments:

(Our responses are in blue)

Reviewer #1 (Remarks to the Author):

The broader context of the paper is understanding arthropod segmentation mechanisms, which are not universal across arthropods, and may contribute to understanding how a homologous process (segmentation) can be achieved in multiple molecular ways. The manuscript by Akaiwa, Oda & Oda is an in-depth single cell genomics characterization of two early embryonic stages of spider development. Though a part of the data has been published earlier for one stage (stage 5, radially symmetric disc), they sequence single-cell and single nuclei for stage 07 (antero-posterior axis patterning), and also single-nuclei for stage 5. The most striking feature of the descriptive part of the paper is the recapitulation of the A-P axis into the 2D UMAP representation of the multidimension data, which is based on expression levels of each cell/nucleus.

The paper can be divided in two components:

- (1) description of the single cell data for both stages, with variation in multiple parameter, validation and cluster annotation with in situ hybridization.
- (2) Method development and genome-wide expression pattern prediction.

This is a beautifully done study. Though a few single-cell datasets have been published for spiders, this study pushes the boundary in that it provides unparalleled resolution and genome-wide biological insights for the question of segmentation.

The characterization of the dataset is exemplary, and I appreciate the exploration of multiple parameters for the UMAPS to show that the head/trunk/tail alignment in the UMAP is robust. The resolution of the data is impressive, and the clusters, even the ones composed of few cells, are well annotated with marker genes, at least in the single-nuclei data. They present and thoroughly characterize the ectoderm cluster, showing that specific regions along the A-P axis of the embryo are represented by nuclei along a line in the UMAP ectodermal cluster. Given that the UMAP reflects the AP axis very well (unusual), they devise a method to approximate a line along the clusters and quantify expression patterns along the line. Even

though the method they developed might not work in the context of other datasets it worked very well for these early embryos, allowing for the discovery of candidate A-P genes. This method recapitulated benchmark genes, and allowed, via clustering, identifying genes with similar patterns. They find that around 200 genes are sufficient to encode the AP information, though data resampling analyses.

I strongly recommend publication: This a great dataset, a show-case of the power of genomics in lifting the rigor of research in emerging model organisms, and foundational for the question of segmentation in arthropods. I have mostly minor comments and requests for clarification.

General comment:

-Color and dot sizes in the UMAPS. The black UMAPS with small dot sizes are very difficult to visualize (ex: Fig 1D, center). I imagine the choice was made to draw a correspondence with the colors in the FISH. If the authors prefer the black background, they could consider increasing dot sizes to improve visualization, and using a lighter gray to show the shape of the clusters better. Alternatively, the UMAPS could all be shown with white background, as the colors would stand out much more clearly.

In response to the reviewer's suggestion, we changed the dot sizes into a larger one in all the UMAP plots, including those with a white background. Moreover, we used a lighter gray for dots representing no gene expression in the black background UMAPs.

Minor comments:

Introduction

The introduction is excellent.

Line 32: I suggest changing for "may be remarkably diverse", because there are certainly cases where it is conserved.

This has been changed, as suggested (line 33).

'the developmental processes that generate these features may be remarkably diverse'

Line 65: I think genome assembly is more accurate.

Corrected, as suggested (line 66). 'With a well-annotated, chromosome-level genome assembly'

Results and Discussion

Line 130–133: Typos in nucleus

We have corrected the typos (line 131–134).

Line 134–135: Could you please briefly elaborate in the text why they may be pseudo clusters?

Such as why would all clusters are expected to have roughly the same number of expressed genes.

We considered these clusters to be pseudoclusters because they exhibited low gene and molecule counts per nucleus and because they lacked distinct cluster-specific markers. We have added a sentence clarifying the latter reasoning regarding these clusters (line 135–138).

‘Furthermore, cluster markers for these and derived clusters were not specifically expressed, showing higher expression levels in other clusters or very faint expression in the respective clusters (Fig. 2, clusters 3, 13, and 15; Supplementary Fig. 4C).’

Figure S3: It would be helpful to add the gene names under the LOC number, in addition to their mention in the figure legend.

As suggested, we have added the gene names to Supplementary figure 4 (previous Fig. S3), as well as to Fig. 3 and Supplementary figure 8 for known genes.

Line 154: It would be helpful to mention in the text (in parenthesis) what the markers for each tissue are called.

We have added the names of the markers (line 156, 174, 298). Additionally, for clarity, Table 2, showing the gene list used for cell and tissue markers, was reproduced from Supplementary table 2.

Lines 151–158: It took me a few passes to understand what was meant here, about the split on not-split cluster to the endo/ex cluster in 1D and 1E. Could you please clarify this section? I also did not understand how the presented data support of not the hypothesis about the origin of the extra embryonic cells. In this passage, I think it would be useful to add a sentence about what is the implication of this difference between nucleus or cell data, or at least what difference we should expect between both datasets.

This section outlines our initial attempt at cluster identification. Therefore, we rewrote the sentence to explain that an ex/endo marker (*At_eW_012_A08*) marked one cluster in the single-cell UMAP and two clusters in the single-nucleus UMAP (line 156–158).

‘Using an endoderm/extraembryonic cell marker (*At_eW_012_A08*), two superclusters were identified in the single-nucleus sample (Fig. 1D) and one cluster in the single-cell sample (Fig. 1E).’

Line 165: Should it be the third graph?

We referred to two figures here, which was ambiguous. Therefore, we have rewritten this part (line 167). ‘(Fig. 1I, the third plot and Fig. 2)’

Line 169: In Figure 3A, some locus projections are not restricted to only the clusters pointed by the arrows. For example, LOC107451631. Here I think it is worthy at least mentioning this, because when stating "perfectly" in this sentence one wonders if expression in the other clusters is real or not.

Regarding the expression of LOC107451631, we detected two expression domains in both the UMAP and embryo at comparable positions. We have added a sentence to explain this (line 173–174). ‘This included the characteristic expression pattern of LOC107451631 (*Pt-osr-B*) in the two distinct domains.’

Additionally, for a clearer comparison, we added arrowheads in Figure 3A to indicate the positions of expression domains in the stained embryos.

Lines 189–191: I think this sentence is circular.

We have rewritten the sentence (line 194–195). ‘the stage-7_cell UMAP plot exhibited a less organized pattern of cell populations expressing the AP markers’

Line 312: The in silico experiments of modifying expression matrices are fascinating. My suggestion for here and the previous instances is to open the paragraph with a clearer explanation of what the goal of the experiment is, instead of "to further investigate their contribution". In this paragraph and before, it would help a lot the reader in following the experiment. So in this case, something as: To investigate the if the structure of the recovered AP axis is dependent on a global vs local contribution of individual clusters, we..."

We added a phrase in accordance with the reviewer’s suggestion (line 318–319).

‘To investigate whether the structure of the recovered AP axis is dependent on the global or local contribution of each cluster,’

Line 369: trough?

We changed the word to ‘between’ (line 378). ‘between the *Pt-hh* bi-splitting peaks’

Line421: selected ectoderm cluster

We added ‘cluster’ (line 441). ‘within the selected ectoderm cluster’

Line 618: remove "that"

Removed, as suggested (line 655). ‘Nuclei located near the anterior end’

Additional comments:

-Are there genes that are repeated between sets A, B and C? B29 and A23 seem to have some shared genes, for example.

We found that two or three genes were shared across several groups. We have added sentences to explain this (line 383–392). ‘Among the groups mentioned here, we noted that some shared two or three genes; -----’

-It would be beneficial to explain what the first derivative is somewhere in the Results or Methods.

We have added a sentence explaining the first derivative (line 335–337), as suggested by the reviewer. ‘The values of the first derivative of a graph correspond to the rate of change, which directly dictates the form of the graph.’

Reviewer #2 (Remarks to the Author):

Summary:

This manuscript presents a comprehensive single nucleus and single cell RNA-seq analysis of segmentation in the spider *Parasteatoda tepidariorum*. This is the third embryonic single-cell sequencing project for this species in the last couple of years, but it focuses on a different subset of developmental stages and asks different questions. The manuscript offers genome-wide, spatially resolved insights into anterior-posterior (AP) patterning. The authors demonstrate that ectodermal AP polarity and segmental stripe formation can be reconstructed from transcriptomic data alone, enabling a novel approach to studying developmental diversity in arthropods. This work represents a significant contribution to evolutionary developmental biology and single-cell genomics. I have no significant criticisms of the manuscript, but only a few technical comments and suggestions for improvement.

General comments:

The readers could benefit from a very brief discussion of the pros and cons of sn/sc use.

We have added sentences describing the differences between our snRNA-seq and scRNA-seq results (line 465–472). ‘One major drawback of our scRNA-seq procedure, -----’

Figures presenting UMAPs with a black background are very difficult to see except at high magnification. It is not clear why a black background was chosen for these figures. If the authors wish to differentiate these panels (highlighting expression of specific genes within the

UMAP), I suggest using a different colored panel outline or something similar, rather than a black background.

We increased the dot size and used a lighter gray color, as explained in our response to Reviewer 1's comment on the same point.

Using "stage 5l" is confusing due to the similarity of a lower-case l and the numeral 1 in different fonts. Indeed, the age in hours of Stage 7 is 51 hours, causing additional confusion. I suggest using "5late" or a similar annotation.

We changed 'stage 5l' to 'late stage 5'.

Specific comments:

Line 46-47 - the sentence refers to differences in genome size and genome content, but the cited references refer to gene content and to whole genome duplications in spiders. Discussion of genome size evolution usually refer to a more continuous change in genome size, and not specifically to duplications. I suggest omitting "genome size" in this context.

We omitted 'genome size' as suggested (line 48). 'variations in gene content'

Line 73 - "bifurcating cell trajectories" - why only bifurcating?

We have changed this phrase to just 'trajectories' (line 74). 'the trajectories of cell differentiation'

Line 252 - "genome widely predict" sounds awkward. I suggest changing to "predict in a genome wide fashion" or something similar.

We have changed the phrase to 'to predict gene expression profiles in a genome-wide and quantitative manner' as suggested (line 258).

Ariel Chipman

Reviewer #3 (Remarks to the Author):

The authors used single-cell and single-nucleus RNAseq data to reconstruct the emergence of AP polarity and segmentation in the common house spider *Parasteatoda tepidariorum*. The careful data analysis reaches an impressive degree of spatial refinement yielding UMAPs with ectodermal clusters corresponding to individual segment analgen. The data set also uncovers new mesodermal and endodermal clusters. Together these data will be the foundation of future

functional studies which promise substantial new insights into spider segmentation and germ layer development. The paper is well written and should be of interest to a broad readership of researchers interested in single cell approaches in general and arthropod segmentation.

Reviewer #4 (Remarks to the Author):

The authors of the manuscript titled “Genome-wide quantitative dissection of an arthropod segmented body plan at single-cell resolution” presented a single cell transcriptome study for the two developmental stages that related to AP axis segmentation emergence. The authors separately analyzed the scRNA-seq/snRNA-seq datasets related to their projects, discovered and validated many embryo body segments specific marker genes. Their studies could be valuable for the developmental study of spiders. However, I have several concerns regarding their current data analyses, which are listed as followed.

(1) The authors mentioned that the two stages chosen for their study represents a critical period of emergence of segmental units along the AP axis. Although the author presented analyses for each stage embryos separately as shown in Figure 1C-E, however, it might be better to directly do an integrated analysis among datasets, which could align the cells across developmental stages.

We agree on the importance of integrated analysis. However, we believe that this analysis requires careful execution due to the fact that the data originated from eggs laid by different female spiders. Furthermore, there were significant time differences (more than 12 hours) between the two stages (late stage 5 and stage 7). This time difference is substantial when we consider the speed of the spider embryogenesis. We propose that data integration be the focus of our next study, in which we aim to perform this analysis using sibling embryos from the same egg sac, collected at regular intervals.

(2) Why the snRNA-seq used in Figure 1G captured more genes than the scRNA-seq used in Figure 1H? As my understanding, snRNA-seq technology normally capture less gene transcripts than scRNA-seq.

As the reviewer noted, scRNA-seq generally captures more genes than snRNA-seq. However, our study detected fewer total genes in the scRNA-seq data compared to the snRNA-seq data, as shown in Table 1. We attribute this to the significantly lower cell recovery in the single-cell library. We have added sentences explaining this discrepancy in the Discussion section (line 465–472).

‘One major drawback of our scRNA-seq procedure, when using stage-7 spider embryos-----’

(3) As the authors mentioned in the method part (line 557-571), the authors only considered the cells with more than 1500 unique genes (more than 2000 genes for snRNA-seq). Could these criteria too stringent for single-cell transcriptome study? Based on previous published single-cell transcriptomic studies, the cells with unique genes between 200 and 1500 still can be useful. Could the authors provide more detail about the data before and after filtering? Such as the number of cells before and after filtering and the distribution of the number of genes before and after filtering.

Following the reviewer’s suggestion, we added a new figure (Supplementary figure 2) showing the number of genes and molecules per cell and nucleus before and after filtering. We also included the number of filtered cells and nuclei in the Methods section (line 593–596).

‘In the filtering, we removed 681 nuclei from the late stage-5_nucleus data -----’

Supplementary figure 2 presents the results of Seurat clustering using the unfiltered counts. These results indicate that the nuclei filtered in our analysis clustered with other low-count nuclei, suggesting that our filtering criteria were not overly stringent. We have added sentences to explain these situations (line 603–607).

‘Additionally, we performed Seurat analyses on the pre-filtered dataset and -----’

Supplementary Figure 2

Supplementary Figure 2. Clustering analyses of the unfiltered cell/nuclei dataset. A–C Violin plots showing the number of genes (upper panels) and molecules (lower panels) detected in each cell or nucleus before (left column) and after (boxed in gray) filtering, UMAP plots showing results of the clustering of the unfiltered datasets (third column), and violin plots for the counts in each cluster shown in the UMAP plots (right) in the late stage-5_nucleus (A), stage-7_nucleus (B), and stage-7_cell (C) datasets. In the UMAP plots, the positions of nuclei or cells that were filtered out are visualized by black dots.